# Robust Losses for Learning Value Functions

## Abstract

Most value function learning algorithms in reinforcement learning are based on the mean squared (projected) Bellman error. However, squared errors are known to be sensitive to outliers, both skewing the solution of the objective and resulting in high-magnitude and high-variance gradients. Typical strategies to control these high-magnitude updates in RL involve clipping gradients, clipping rewards, rescaling rewards, and clipping errors. Clipping errors is related to using robust losses, like the Huber loss, but as yet no work explicitly formalizes and derives value learning algorithms with robust losses. In this work, we build on recent insights reformulating squared Bellman errors as a saddlepoint optimization problem, and propose a saddlepoint reformulation for a Huber Bellman error and Absolute Bellman error. We show that the resulting solutions have significantly lower error for certain problems and are otherwise comparable, in terms of both absolute and squared value error. We show that the resulting gradient-based algorithms are more robust, for both prediction and control, with less stepsize sensitivity.

## 1 Introduction

Learning value functions from off-policy data remains an open challenge due to high-variance samples and the inability to optimize the objective of interest. Progress towards this goal has been made over years of algorithm development, by reducing the variance for temporal difference (TD) algorithms (Precup et al., 2000; Munos et al., 2016; Mahmood et al., 2017); following approximate gradients of a proxy objective—the mean squared Bellman error (MSBE)—which upper bounds our objective of interest (Dai et al., 2017; 2018; Feng et al., 2019); and following an approximate gradient of a *projected* version of the MSBE (Sutton et al., 2009; Maei et al., 2009; Mahadevan et al., 2014; Liu et al., 2016; Ghiassian et al., 2020). For many years, however, there was only a limited set of choices—mainly vanilla TD algorithms. Because it was unclear precisely what objective TD was optimizing (Baird, 1995; Antos et al., 2008), it was difficult to extend the algorithm. The known alternative—optimizing the MSBE—suffers from an issue of double sampling in the absence of a simulator (Baird, 1995; Scherrer, 2010).

Driving the recent innovations are two key advances for objectives in RL. The first is the formalization of the objective underlying TD, called the mean squared projected Bellman error (MSPBE) (Antos et al., 2008; Sutton et al., 2009), which projects the Bellman error into the space spanned by the function approximator. Many algorithms are built on the originally introduced variants, GTD2 and TD with gradient corrections (TDC) (Sutton et al., 2009). These algorithms, however, are limited to linear function approximation because the MSPBE is defined only for the linear case.

The second advance is the introduction of a conjugate form for the MSBE to handle the double sampling problem (Dai et al., 2017). By transforming the MSBE using biconjugate functions, the double sampling problem instead became a better understood saddlepoint optimization problem. The SBEED algorithm (Dai et al., 2018) later extended the conjugate MSBE to control using a smoothed Bellman optimality operator and parameterizing both the policy and value function estimates. By transforming the MSBE into an objective for which we can obtain unbiased sample gradients, the conjugate MSBE allowed a natural extension to nonlinear function approximation.

A natural next step is to use these advances to revisit defining and optimizing statistically robust losses for value functions. The MSBE and MSPBE are built on squared errors, which are known to be sensitive to outliers. In the RL setting, this translates into overemphasizing states for which the Bellman residual is high, at the cost of obtaining accurate estimates in other states. For example, consider the CliffWorld domain (Sutton and Barto, 2018) where the agent must navigate alongside

a cliff to reach a rewarding goal state. Should the agent step into the cliff, the agent is sent back to the start state with a large negative reward. In order to represent the optimal policy, the agent must learn that actions which lead to the cliff yield negative return, while actions that lead towards the goal yield positive return (or at least *less* negative). Perfectly representing the exact magnitude of negative returns for stepping in the cliff is not useful, as it is sufficient to only know that these actions are more negative than actions leading towards the goal. Further, because the agent bootstraps off these value estimates in the states near the goal, squared errors tend to magnify these inaccuracies across all states visited in the episode. In general, the ability to replace squared errors with alternatives, like absolute errors or the Huber loss (Huber, 1964), provides another dimension to improve our algorithms and, to the best of our knowledge, has yet to be explored.

Issues with squared errors have been noted in the RL literature and addressed heuristically in control using clipping. DQN uses clipped TD errors by default to avoid large magnitude updates (Mnih et al., 2015), likewise, Dueling DQN (Wang et al., 2016) directly clips its gradients. There is a close relationship between clipping TD errors and using a Huber loss in the linear function approximation setting. However, even for linear Q-learning or TD-learning, clipping TD errors does not correspond to minimizing a Huber loss for the Bellman error. Rather, it takes a non-gradient update and applies clipping to that update to mimic the Huber loss which results in a bias similar to residual TD methods (Baird, 1995; Sutton and Barto, 2018).

In this work, we develop practical algorithms that can use absolute and Huber errors instead of squared errors for the BE. To do so, we develop a unified perspective of the mean absolute Bellman Error (MABE), mean squared Bellman Error (MSBE), and mean Huber Bellman Error (MHBE) which smoothly interpolates between these other two. We rely on the insight that the MSBE can be reformulated into a saddlepoint problem with the introduction of an auxiliary learned variable; using the same strategy, we derive a biconjugate form for the MHBE and MABE amenable to simple gradient-descent techniques. The resulting approach is a simple modification of existing gradient TD algorithms, using a different update for the auxiliary variable, making it straightforward to use either of these losses. We show that the MHBE and MABE can significantly change the solution quality, improving accuracy in terms of the mean squared value error, as well as the mean absolute value error. We then show that gradient algorithms for the MHBE tend to perform similar to those for the MSBE, but provide significant robustness improvements in certain cases, particularly under bad state aliasing. Finally, we show that in control with nonlinear function approximation, gradient-based algorithms minimizing the MHBE often outperform those using squared losses or those using non-gradient updates.

## 2 PROBLEM FORMULATION

We model the agent's interactions with its environment as a Markov Decision Process (MDP), $(\mathcal{S}, \mathcal{A}, P, R, \gamma)$. At each time-step $t$, the agent observes states $S_t \in \mathcal{S}$, selects an action $A_t \in \mathcal{A}$ according to policy $\pi : \mathcal{S} \to \Delta(\mathcal{A})$, transitions to the next state $S_{t+1} \in \mathcal{S}$ according to transition function $P : \mathcal{S} \times \mathcal{A} \to \Delta(\mathcal{S})$, and receives a scalar reward signal $R_{t+1}$ and discount $\gamma_{t+1} \in [0, 1]$. The discount depends on the state, and encodes termination when $\gamma_{t+1} = 0$ (White, 2017).

For the prediction setting, the agent's goal is to estimate the value function $v_\pi$ for a given policy. The value function can be defined recursively, using the Bellman operator

$$(\mathcal{T}v)(s) \stackrel{\text{def}}{=} \mathbb{E}[R_{t+1} + \gamma_{t+1}v(S_{t+1}) \mid S_t = s]$$

where the expectation is taken with respect to the policy $\pi$ and transition dynamics $P$. The true values $v_\pi$ are the fixed point for this operator: $\mathcal{T}v_\pi = v_\pi$. Our goal is to approximate $v_\pi$, with $v_\theta \in \mathcal{F}$ for some (parameterized) function space $\mathcal{F}$. Typically, the quality of this approximation is evaluated under the value error, either mean squared value error (MSVE) or mean absolute value error (MAVE)

$$\text{MSVE}(v_\theta) \stackrel{\text{def}}{=} \sum_{s \in \mathcal{S}} d(s)\left(v_\theta(s) - v_\pi(s)\right)^2 \qquad \text{MAVE}(v_\theta) \stackrel{\text{def}}{=} \sum_{s \in \mathcal{S}} d(s)|V(s) - v_\pi(s)|$$

where $d : \mathcal{S} \to [0, 1]$ is typically the visitation distribution under a behavior policy.[1]

---

[1]For exposition, we assume discrete states and actions throughout this paper. The connection to continuous state-actions is straightforward and we will explicitly call out where this connection is less obvious.

One objective used to learn approximation $v_\theta$ is the mean squared Bellman error (MSBE)

$$\text{MSBE}(\theta) \stackrel{\text{def}}{=} \sum_{s \in \mathcal{S}} d(s)((\mathcal{T}v_\theta)(s) - v_\theta(s))^2 = \sum_{s \in \mathcal{S}} d(s)\mathbb{E}[\delta(\theta) \mid S = s]^2 \tag{1}$$

where $\delta(\theta) \stackrel{\text{def}}{=} R_{t+1} + \gamma_{t+1}v_\theta(S_{t+1}) - v_\theta(S_t)$. If $v_\pi \in \mathcal{F}$, then there exists a $\theta$ such that $v_\theta = v_\pi$ and $\text{MSBE}(\theta) = 0$. Otherwise, if $v_\pi \notin \mathcal{F}$, then this objective trades-off Bellman error across states.

The trade-off in errors across states is impacted by both the weighting $d$ as well as the fact that a squared error is used. The function approximator focuses on states with high weighting $d$, which is sensible. However, by using a squared error, it heavily emphasizes states with higher error which may not be desirable. In the next section, we develop an approach to use robust losses—the absolute error and the Huber error—in place of this squared error.

The same approach as above can also be used for control to approximate the optimal action-values $q^*$. These values can similarly be defined using a Bellman optimality operator

$$(\mathcal{T}^*q)(s,a) \stackrel{\text{def}}{=} \mathbb{E}\left[R_{t+1} + \gamma_{t+1}\max_{a' \in \mathcal{A}} q(S_{t+1}, a') \mid S_t = s, A_t = a\right]$$

with $\mathcal{T}^*q^* = q^*$. The corresponding mean squared Bellman error for learning approximate $q_\theta$ is

$$\sum_{s \in \mathcal{S}, a \in \mathcal{A}} d(s,a) \left[(\mathcal{T}^*q_\theta)(s,a) - q_\theta(s,a)\right]^2$$

where we overload $d$ to be a state-action weighting, which typically corresponds to the state-action visitation frequency under a policy.

## 3   ROBUST BELLMAN ERRORS

In this section, we provide reformulations of the MABE and the MHBE which avoid the double sampling issue by using their biconjugates. For some intuition on how these objectives differ from the MSVE and MSBE, we visualize them in Figure 1.

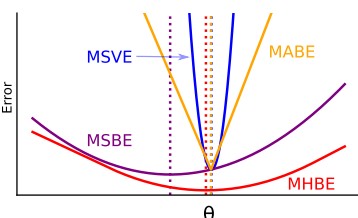

Figure 1: Objective values and fixed-points on the HardAlias-2 MDP (defined in Section 5). The fixed-points of the absolute and Huber objectives are much better proxies for the squared value error than the fixed-point of the squared Bellman error. The dotted vertical lines indicate minima of each objective.

The absolute Bellman error is straightforward to specify for a state, $|\mathbb{E}[\delta(\theta) \mid S = s]|$, as is the Huber Bellman error, $p_\tau(\mathbb{E}[\delta(\theta) \mid S = s])$, where the Huber function is

$$p_\tau(a) \stackrel{\text{def}}{=} \begin{cases} a^2 & \text{if } |a| \leq \tau \\ 2\tau|a| - \tau^2 & \text{otherwise} \end{cases}$$

for some $\tau \geq 0$. A common choice in RL is $\tau = 1.0$, which corresponds to using squared error when the magnitude of the error is below 1, and absolute error otherwise.

The difficulty, however, is obtaining a sample of the gradient of this objective for the same reason as the MSBE: the double sampling problem. To see why, let us examine the gradient of the MSBE

$$\nabla\text{MSBE}(\theta) = \sum_{s \in \mathcal{S}} d(s)\mathbb{E}[\delta(\theta) \mid S = s]\,\nabla\mathbb{E}[\delta(\theta) \mid S = s].$$

To sample this gradient requires a sample of $\delta(\theta)$ for the first expectation and an independent sample of $\delta(\theta)$ for the second expectation. Otherwise, using the same sample, we estimate the gradient of $\mathbb{E}[\delta(\theta)^2 \mid S = s]$ instead of $\mathbb{E}[\delta(\theta) \mid S = s]^2$. Due to the chain rule, both the MABE and MHBE will suffer from the same issue as the MSBE.

The strategy is to reformulate the objectives using conjugates, introducing an auxiliary variable to estimate part of the gradient. For a real-valued function $f : \mathbb{R} \to \mathbb{R}$, the conjugate is $f^*(h) \stackrel{\text{def}}{=} \sup_{x \in \mathbb{R}} xh - f(x)$. This function $f^*$ also has a conjugate, $f^{**}$, which is called the biconjugate of $f$. Further, for any function $f$ that is proper, convex, and lower semi-continuous, the biconjugate $f^{**}(x) = f(x)$ for all $x$ by the Fenchel-Moreau theorem (Fenchel, 1949; Moreau, 1970). Because the three functions we want to reformulate—the absolute, huber, and square functions—are all proper,

convex, and lower semi-continuous, this equivalence allows us to reformulate these losses using biconjugates to avoid the double sampling issue without changing the solutions to these losses.

To rewrite existing results in our notation and provide some intuition, we first write the reformulation for the MSBE. The conjugate of the squared error $f(x) = \frac{1}{2}x^2$ is $f^*(h) = \max_{x \in \mathbb{R}} hx - \frac{1}{2}x^2$, which is in fact again the squared error: $f^*(h) = \frac{1}{2}h^2$ (this result is well known, but for completeness we include the proof in Appendix A). The biconjugate is $f^{**}(x) = \max_{h \in \mathbb{R}} xh - \frac{1}{2}h^2$ and $f(x) = f^{**}(x)$. We can use this to get that, for $x = \mathbb{E}[\delta(\theta) \mid S = s]$,

$$\mathbb{E}[\delta(\theta) \mid S = s]^2 = \max_{h \in \mathbb{R}} 2\mathbb{E}[\delta(\theta) \mid S = s] h - h^2.$$

If we have function space $\mathcal{F}_{\text{all}}$—the set of all possible functions $h : \mathcal{S} \to \mathbb{R}$—then we get that

$$\text{MSBE}(\theta) = \sum_{s \in \mathcal{S}} d(s) \max_{h \in \mathbb{R}} (2\mathbb{E}[\delta(\theta) \mid S = s] h - h^2)$$

$$= \max_{h \in \mathcal{F}_{\text{all}}} \sum_{s \in \mathcal{S}} d(s)(2\mathbb{E}[\delta(\theta) \mid S = s] h(s) - h(s)^2)$$

where the maximization comes out of the sum using the interchangeability property (Shapiro et al., 2014; Dai et al., 2017) and $h(s)$ is a function that allows us to independently pick a maximizer for every state in the summation.

If we have the maximizing function $h^*(s)$, it is straightforward to sample the gradient. Because $h^*(s)$ itself is not directly a function of $\theta$, then the gradient is

$$\nabla_\theta \sum_{s \in \mathcal{S}} d(s)(2\mathbb{E}[\delta(\theta) \mid S_t = s] h^*(s) - h^*(s)^2) = \sum_{s \in \mathcal{S}} 2d(s)h^*(s)\mathbb{E}[\gamma \nabla_\theta v(S') - \nabla_\theta v(s)) \mid S_t = s].$$

Drawing samples $S \sim d(\cdot)$, $A \sim \pi(\cdot|S)$, and $S' \sim P(\cdot|S, A)$, we can easily compute a stochastic sample of the gradient. In practice, we simply optimize the resulting saddlepoint problem with a minimization over $\theta$ and maximization over $h$. Note the optimal $h^*(s) = \mathbb{E}[\delta(\theta) \mid S_t = s]$.

We can use the same procedure for the Huber error and the absolute error. We derive the biconjugate form for the Huber error in the following proposition. Though it is a relatively straightforward result to obtain, to the best of our knowledge, it is new and so worth providing formally.

**Proposition 3.1** *The biconjugate of the Huber function is*

$$p_\tau^{**}(x) = \max_{h \in [-\tau, \tau]} xh - \frac{1}{2}h^2. \tag{2}$$

Due to space restrictions, we provide a complete proof in Appendix A.

The absolute value has biconjugate $\max_{h \in [-1,1]} xh$. As in the squared error case, this is a well known result but we include the proof for completeness in Appendix A. Notice again the constrained optimization problem for this biconjugate, as in the case of the Huber biconjugate function.

We can now provide the forms for MABE and MHBE:

$$\text{MABE}(\theta) \stackrel{\text{def}}{=} \max_{h \in \mathcal{F}_{\text{sign}}} \sum_{s \in \mathcal{S}} d(s)h(s)\mathbb{E}[\delta(\theta)|S = s]$$

$$\text{MHBE}(\theta) \stackrel{\text{def}}{=} \max_{h \in \mathcal{F}_{\text{clip}_\tau}} \sum_{s \in \mathcal{S}} d(s)(2h(s)\mathbb{E}[\delta(\theta)|S = s] - h(s)^2)$$

$\mathcal{F}_{\text{sign}}$ is the set of all functions $h_{\text{sign}} : \mathcal{S} \to \{-1, 1\}$ and $\mathcal{F}_{\text{clip}_\tau}$ the set of all functions $h_{\text{clip}_\tau} : \mathcal{S} \to [-\tau, \tau]$.

For all of these reformulations, in practice we will have parameterized functions $h$, and so only approximate the objective. For example, for the MSBE, we may use linear functions $\mathcal{F}_h = \{h \mid h(s) = \theta_h^\top x(s), \theta_h \in \mathbb{R}^d\}$ where $x : \mathcal{S} \to \mathbb{R}^d$ is a $d$-dimensional feature generating function. For the MHBE, we might use $\mathcal{F}_h = \{h \mid h(s) = \text{clip}_\tau\left(\theta_h^\top x(s)\right), \theta_h \in \mathbb{R}^d\}$.

For the conjugate reformulation of the MSBE, limiting $v_\theta$ and $h(s)$ to both be linear functions of the same features results in the mean squared Projected Bellman error, as noted by Dai et al. (2017).

There are no equivalent existing projected Bellman errors for these new absolute and Huber variants. We note, however, that when the function space is constrained to only functions of our features $x(s)$, these conjugate reformulations do not suffer from the same identifiability issue raised for the MSBE (Sutton and Barto, 2018; Scherrer, 2010) as we show in Appendix B.

The final form of the MHBE highlights a connection to a recently proposed gradient TD algorithm that seemed to significantly improve stability over vanilla gradient TD methods. The algorithm, called TD with Regularized Corrections (TDRC), adds a regularizer to the parameters for $h$ (Ghiassian et al., 2020). This regularizer has the effect of constraining $h(s)$ to be closer to zero, and so could be seen as providing some of the same robustness benefits as a Huber loss. Of course, the correspondence is by no means exact. Regularizing with $\ell_2$ on the parameters is different than restricting between $-1$ and $1$. Further, as shown in that work, TDRC does not alter the fixed-point of its mean squared objective. Using the Huber error in place of the squared error, however, will likely alter the fixed-point.

Finally, we show that the MHBE bounds the MAVE. It is well-known that absolute errors are hard to optimize and so the Huber error acts as a smooth approximation to the absolute error. Theorem 3.2 first shows that minimizing the MABE acts as a proxy for minimizing the MAVE. Then, by bounding the absolute value error with the Huber error, we get that the MHBE acts as an approximation of the MABE implying that minimizing the MHBE likewise minimizes the MAVE. Because of this approximation, the MHBE has an irreducible error term which is controlled by the Huber parameter $\tau$. We provide a complete proof of Theorem 3.2 in Appendix A.1.

**Theorem 3.2** *Let* $\tau_{cap} = \min(\tau, 1)$, *then for arbitrary* $v \in \mathbb{R}^d$ *and* $0 < \epsilon < \tau_{cap}^2$ *we have*

$$\|v_\pi - v\|_1 \le \|(I - P_{\pi,\gamma})^{-1}\|_1 \|\mathcal{T}v - v\|_1 \le \|(I - P_{\pi,\gamma})^{-1}\|_1 \sum_{s=0}^{d} \left( \frac{\sqrt{\epsilon}}{2\epsilon} p_\tau \left( \mathcal{T}v_s - v_s \right) + \frac{\sqrt{\epsilon}}{2} \right).$$

## 4 OPTIMIZING THE OBJECTIVES

The gradient of all the reformulated objectives in terms of $\theta$, for a given $h$, is actually the same: $\sum_{s \in \mathcal{S}} d(s)h(s)\mathbb{E}[\nabla \delta(\theta) \,|\, S = s]$. This means that the job of selecting between the absolute, squared, and Huber Bellman errors rests solely on how we approximate the secondary variable, $h(s)$. There are many ways to estimate the $h$ for the MHBE and MABE. A natural starting point would be to use the same estimate, $\tilde{h}_{\theta_h}(s) \approx \mathbb{E}[\delta \,|\, S = s]$, as in the mean squared case, then apply the corresponding non-linear function to $\tilde{h}$—sign or clipping. This gives the following updates for the objectives.

▷ MABE

$$h(s_t) = \text{sign}(\tilde{h}_{\theta_h,t}(s_t))$$

▷ MHBE

$$h(s_t) = \text{clip}_\tau \left( \tilde{h}_{\theta_h,t}(s_t) \right)$$

▷ MSBE

$$h(s_t) = \tilde{h}_{\theta_h,t}(s_t)$$

$$\theta_{h,t+1} = \theta_{h,t} + \alpha_h \left( \delta_t - \tilde{h}_{\theta_h,t}(s_t) \right) \nabla_{\theta_h,t} \tilde{h}_{\theta_h,t}(s_t) \tag{3}$$

$$\theta_{t+1} = \theta_t + \alpha_v h(s_t) \left( \nabla_{\theta_t} v(s_t) - \gamma_{t+1} \nabla_{\theta_t} v(S_{t+1}) \right) \tag{4}$$

Notice if we specifically parameterize $\tilde{h}_{\theta_h}(s) = \theta_h^\top x(s)$ and $v(s) = \theta^\top x(s)$, then we recover the GTD2 algorithm with linear function approximation (Sutton et al., 2009). Because the update for the primary weights is exactly the same as GTD2 and because the clip function encodes box-constraints on the secondary weights (and so is closed and convex), convergence of the GTD2-like algorithm for the MHBE follows directly from Nemirovski et al. (2009).

However, this parameterization is likely not ideal for either MABE or MHBE, because the function approximation does not find the best $h$ under the constraints. Another strategy is to pick a parameterized function class that encodes the constraints. For instance, in the case of the MABE the function, $h(s)$ is effectively a 2-class classifier for the sign of $\mathbb{E}[\delta \mid S = s]$, which can be easily parameterized using a logistic regressor for example. Likewise, for the MHBE, the clipping function can be approximated using a rescaled logistic function $\text{clip}_\tau (x) \approx \tau \tanh(\frac{1}{\tau}x)$, which we will use for the nonlinear control algorithm in Section 6. Despite being approximations, directly parameterizing the sign and clip functions may provide some advantages such as being a smooth approximation to the sign function or a twice differentiable approximation to the clip function; improving their optimization surface.

Several past works have reported a large performance difference between the GTD2 saddlepoint algorithm and the TDC gradient-correction algorithm (White and White, 2016; Ghiassian et al., 2018). The primary parameter vector in gradient-correction methods depends directly on a sample of the TD error, $\delta$, thus benefits from a direct unbiased error signal. Assuming linear function approximation, the update is $\delta x(s_t) - \gamma_{t+1} h(s_t) x(s_{t+1})$. The saddlepoint algorithms, on the other hand, rely fully on $h$—as in the updates above—providing a low-variance but possibly highly biased update.

Manipulating GTD2 update shows the relationship between the saddlepoint and gradient-correction

$$
\begin{aligned}
-\nabla_\theta \delta(\theta) h(s_t) &= h(s_t)\nabla_\theta v(s_t) - \gamma h(s_t)\nabla_\theta v(S_{t+1}) \\
&= (h(s_t) - \delta(\theta) + \delta(\theta))\nabla_\theta v(s_t) - \gamma h(s_t)\nabla_\theta v(S_{t+1}) \\
&= \underbrace{\delta(\theta)\nabla_\theta v(s_t) - \gamma h(s_t)\nabla_\theta v(S')}_{\text{TDC update}} + \underbrace{(h(s_t) - \delta(\theta))\nabla_\theta v(s_t)}_{\text{extra term}}
\end{aligned}
$$

where we end up with a gradient that looks similar to the update for TDC, but with an extra term that accounts for the deviation between $\delta$ and our estimate, $h(s)$. In fact, in the case of linear function approximation, and when $h$ is the best linear approximation, the expected value of this additional term over all states is zero, making TDC an unbiased estimate of the true gradient as shown in Appendix C.

Similarly, we can construct a corresponding gradient correction update to optimize the robust objectives, but at the cost of some additional bias in the approximate gradient because the extra term does not go away in expectation for the MHBE or MABE. As in the original TDC algorithm, the update for the secondary parameters $\theta_h$ remains the same.

$$
\theta_{t+1} = \theta_t + \alpha_v \left( \delta(s_t)\nabla_{\theta_t} v(s_t) - \gamma_{t+1} h(s_t)\nabla_{\theta_t} v(S_{t+1}) \right)
$$

We report the bias of the gradient for this update, for both the MABE and MHBE, in Appendix C. We empirically investigate both updates for all three objectives in Section 5.

## 5 PREDICTION EXPERIMENTS

We first investigate the quality of the fixed-points of each objective on several linear prediction tasks. The goal is to demonstrate the large advantage of the robust objectives in some settings, while maintaining reasonable performance across settings. We then investigate two families of online stochastic algorithms to optimize these objectives. We demonstrate that the Huber objective can often improve—and never harms—the optimization procedure for online learning.

**Environments**

We investigate six different problem settings, each chosen to highlight particular challenges for each objective. The first two use a challenging state representation with features that aggressively alias across states. The robust objectives perform favorably on these problem settings, while the MSBE finds solutions that tradeoff accuracy in the aliased states poorly. **HardAlias-1** is an 8-state random walk where the first, third, and final states share a common feature, and the remaining five states share three features. **HardAlias-2** is the 2-state problem from Tsitsiklis and Van Roy (1997), which was originally designed to highlight the insufficiency of minimizing the squared Bellman Residual. We lightly modify the reward function of the MDP so the optimal value function cannot be perfectly represented allowing each objective to have different minima.

The next investigated problem setting (**Outlier**) is designed to highlight the advantages of the MHBE by creating a single outlier state with a large magnitude return among a large set of states with approximately normally distributed returns. To emulate a more realistic learning scenario, we use a randomly initialized frozen neural network to generate five features to describe 50 state. The agent starts in a state that has an $\epsilon = 0.01$ chance of terminating immediately with -1000 reward, or a $1 - \epsilon$ chance of entering the middle state of a 49-state random walk.

The next pair of investigated problems are chosen to highlight a scenario where the MSBE finds favorable solutions compared to the robust objectives. In these problem settings, the returns are distributed approximately normally across states and states are lightly aliased. We use two random walks, the first with $N = 5$ states (**SmallChain**) and the other with $N = 19$ states (**BigChain**), with a randomly initialized neural network representation of size $\frac{N}{2}$. The agent receives a reward of $-1$ or $+1$ on the left and right-most states respectively.

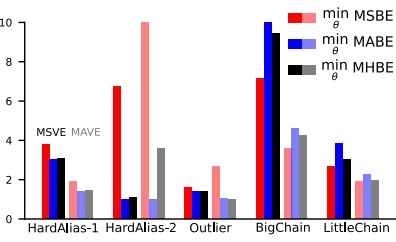

Figure 2: Evaluating the quality of the fixed-points of each objective function according to the MSVE and MAVE across several prediction problems. Error is plotted relative to the best representable value function. The robust losses are better in the hard aliasing domains, the MHBE is slightly better in Outlier, and the MSBE is better on the classic random walks. **Note:** the error for the MSBE is clipped in HardAlias-2 (approx. 25 MAVE).

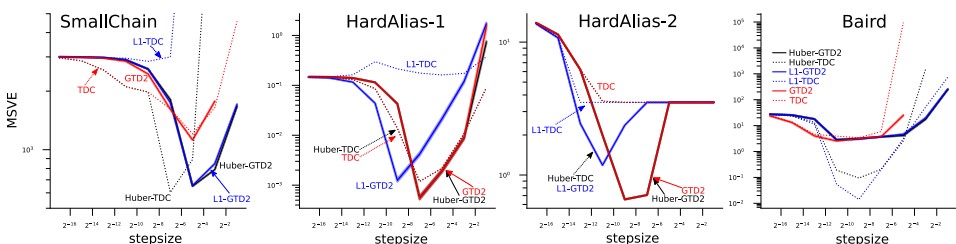

Figure 3: MSVE averaged over 100 independent trials, for each stepsize in prediction domains. The mean squared algorithms generally performed well across environments—even the adversarially chosen environments—suggesting the difficulty in minimizing the MABE. The Huber algorithms performed best across many environments, often displaying less sensitivity to the choice of stepsize.

The final investigated problem setting (**Baird**) is the well-known star MDP from Baird (1999). We use this domain to investigate optimization performance in a high-variance off-policy setting where TD diverges. Because the true values are representable, the fixed-points of each objective are equivalent.

**Investigating the fixed-points**

In this section, we seek to understand the quality of the *solution* of each proposed objective function according to the MSVE and MAVE metrics. We first assume access to a perfect model of each problem setting, then analytically compute the batch gradient for each objective and perform gradient descent using the ADAM optimizer (Kingma and Ba, 2015) and iterate averaging. We additionally assume access to the true underlying state for each MDP and that partial observability due to state aliasing occurs from poor feature selection. This setting allows us to understand the properties of our feature generating functions, our MDPs, and our policies for which certain losses—such as the MSBE—may suffer. For problem settings with randomly initialized feature representations, we perform this procedure for 1000 randomly generated representations and report the average error.

Figure 2 shows the relative MSVE and MAVE for each fixed-point across problems. For MSVE, the robust objective fixed-points are consistently either the best or only off by a comparatively small margin from the squared objective fixed-points. The squared objective fixed-points, however, can have catastrophically bad performance, even on small problems like HardAlias-2.

**Investigating the optimization algorithms**

We investigate the effectiveness of the optimization algorithms proposed in Section 4 on a subset of the linear prediction problem settings. We choose one representative problem setting that favored the MABE and one that favored the MSBE when evaluating the fixed-points. We perform ten thousand update steps for every algorithm on each problem, except on HardAlias-1 where we use one hundred thousand to ensure learning has slowed for every algorithm.

We look at performance across a wide range of stepsizes, $\alpha \in \{2^{-16}, 2^{-15}, \ldots, 2^{-1}\}$, and report the mean and standard error over 100 independent trials for each stepsize, algorithm, and problem setting. To choose the secondary stepsize for each algorithm, we sweep over ratios $\eta = \frac{\alpha}{\alpha_v} \in \{2^{-4}, 2^{-3}, \ldots, 2^3, 2^4\}$ and report performance using the best secondary stepsize for every point.

In Figure 3 we show stepsize sensitivities for each problem and algorithm measured using the MSVE. Due to the similarity in conclusions, we relegate the MAVE sensitivities to Appendix D. The mean absolute TDC variant appears to suffer as a result of biased gradient estimates and generally performs worse across stepsizes than its GTD2-based counterpart. Generally the robust GTD2 algorithms and the Huber TDC algorithm show wider stepsize sensitivities and often the MHBE algorithms show marginally better performance for their best choice of stepsize.

## 6 Nonlinear Control Experiments

Our experiments so far focused on prediction with linear function approximation. However, one of the primary motivating factors of using conjugate Bellman errors is the natural and theoretically sound extension to nonlinear function approximation and control. In this section, we empirically investigate a Huber algorithm for nonlinear control, where we estimate $q_\theta$ and $h$ using neural networks.

**The QRC-Huber Algorithm**

To estimate the secondary parameters of the MHBE for control, we use a two-headed neural network where each head has one output for every action. The first head estimates $q_\theta(s, a)$ and the second head estimates $\tilde{h}(s, a)$. We block gradients from being passed back from the second head of the network, allowing the network's full function approximation resources to be used for predicting $q_\theta(s, a)$ as accurately as possible. This parameterization was used for an algorithm called QRC, an extension of the TDRC algorithm to control (Ghiassian et al., 2020). As discussed in Ghiassian et al. (2020)—and reconfirmed in our own experiments in Appendix D—using the saddlepoint update rule leads to poor performance in control, so we choose to use the gradient correction update.

This results in the following update rules

$$\theta_{h,t+1} = \theta_{h,t} + \alpha \left( \delta - \tilde{h}(s, a) \right) \nabla_{\theta_{h,t}} \tilde{h}(s, a) - \alpha\beta\theta_{h,t}$$

$$\theta_{t+1} = \theta_t + \alpha(\delta\nabla_\theta Q(s, a) - \gamma h(s, a)\nabla_\theta \max_a Q(S', a))$$

where $\theta$ refers to all of the parameters of the neural network, except the parameters for the secondary head, and $\beta$ is the regularization parameter from Ghiassian et al. (2020). Unlike in the prediction setting, we choose to use a twice differentiable approximation of the clipping function to allow easier optimization with pseudo-second order methods like ADAM (Kingma and Ba, 2015). We accomplish this using the tanh function $h(s, a) = \tau \tanh(\frac{1}{\tau}\tilde{h}(s, a))$.

**Experiments in Classic Control Domains**

For the nonlinear control experiments, we investigate three classic control problems—Mountain Car, Cart-pole, and Acrobot—from the Gym suite (Brockman et al., 2016), a larger domain with a heavily shaped reward—Lunar Lander—and one additional domain designed to be particularly challenging for squared error algorithms, Cliff World. For all domains, discount factor $\gamma = 0.99$ and $\epsilon = 0.1$ for the $\epsilon$-greedy policy. The episode is cutoff if the agent fails to reach a terminal state in a pre-specified number of steps. When cut off, the agent is teleported back to the start state and does not update its value function, thus preventing the agent from bootstrapping over the teleportation transition.

For all environments and algorithms, we sweep a broad range of stepsizes and report results for every swept stepsize in Appendix D.. For QRC-Huber, we fix the Huber threshold parameter $\tau = 1$ for all domains except Mountain Car, where we use $\tau = 2$. We further ablate the impact of this decision in Appendix D. For the QRC methods, we chose not to use target networks—a frozen, infrequently updated set of weights for the bootstrapping target—so that we can highlight the stability provided by using true gradient-based methods with robust losses. DQN uses targets networks and sweeps over multiple refresh rates and additionally sweeps over its clipping parameter. In total, QRC and QRC-Huber tune over 6 meta-parameter combinations while DQN tunes over 120.

To demonstrate the stability of each algorithm, we report the full distribution of the performance metric over 100 independent trials for the best stepsize on each domain. We use the average return achieved over the last 25% of steps as our performance metric. We expect algorithms which exhibit stable performance to have a narrow, approximately normal distribution centered around higher return, while algorithms which are unstable we expect to have wide performance distributions or even multi-modal distributions. We also report standard learning curves, in Figure 5.

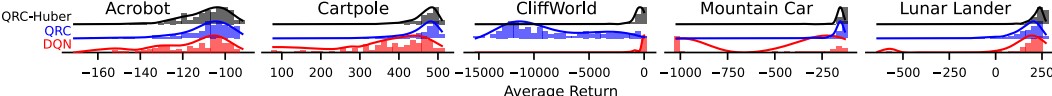

Figure 4: Subplots show the distribution over 100 random seeds. The performance measure is the average return over the last 25% of steps for the best stepsize meta-parameter chosen per-domain. QRC-Huber consistently has approximately normal and narrow distributions around high-performance returns. DQN has highly inconsistent behavior over random seeds, with bimodal performance on Mountain Car and Lunar Lander, and very long-tailed performance on Acrobot and Cartpole.

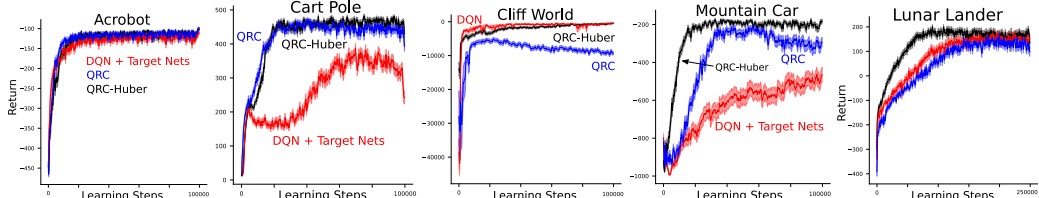

Figure 5: Learning curves for the best meta-parameter configuration for each domain, averaged over 100 random seeds. Shaded regions indicate one standard error. In Acrobot and Cart Pole, QRC-Huber and QRC have similar performance. In Acrobot and Cliff World, DQN and QRC-Huber have similar performance. However, in Mountain Car and Lunar Lander, QRC-Huber has significantly better performance than both competitors. QRC-Huber is the only algorithm to reliably solve all domains.

Figure 4 shows the performance distributions of each tested algorithm. QRC-Huber exhibits narrow and approximately normal performance distributions for every domain, suggesting the stability of the algorithm over random seeds. The QRC algorithm performs reasonably on the Acrobot and Cart-pole domains, but performs quite poorly on the Cliff World domain. Because QRC is based on the mean squared Bellman error, the poor performance on Cliff World is exactly as expected, since this domain was chosen adversarially to highlight challenges with mean squared errors. While DQN is based on a clipped loss function that appears similar to the mean Huber Bellman error, it does not seem to enjoy the same stability as QRC-Huber, with average performance far worse than QRC-Huber on four of five domains due to high bimodality or long-tailed performance distributions. The learning curves in Figure 5 further highlight that QRC-Huber is the most robust of the three, across all five problem setting, either having comparable or notably better performance.

**Experiments on Minatar**

Finally, we demonstrate that QRC-Huber can scale to larger domains using more complex neural network architectures. We use the Minatar suite of five miniaturized Atari games which retain much of the complexity of the full Atari games, while considerably reducing the computational requirements and cost (Young and Tian, 2019). We allow all three control algorithms to sweep over a small range of stepsizes and allow only DQN to sweep over target network refresh rates, as both QRC and QRC-Huber do not require target networks. We set the discount factor $\gamma = 0.99$ for all domains and use the same neural network architecture and default parameters as Young and Tian (2019).

To avoid domain overfitting and reduce the cost of meta-parameter tuning, we treat the entire Minatar suite as a single problem setting. As such, each algorithm must pick one meta-parameter setting to use across all five games; favoring algorithms which are stable and insensitive to parameter choices. We scale the expected returns from each domain using probabilistic performance profiles (Jordan et al., 2020; Barreto et al., 2010), then report the average scaled performance across the entire suite with 95% confidence intervals. We run each algorithm with its best meta-parameter setting for 30 runs on each domain for a total of 150 runs. Additional procedural details can be found in Appendix E.

Table 1: Average return on Minatar

| | |
|---|---|
| QRC-Huber | $0.53 \pm 0.03$ |
| QRC | $0.47 \pm 0.02$ |
| DQN | $0.36 \pm 0.06$ |

In Table 1, we report the average scaled return across games in the Minatar suite. QRC-Huber outperforms both QRC and DQN on average across domains. Despite having four times the number of meta-parameter combinations and the ability to use target networks, DQN performs considerably worse than either gradient-based algorithm. That QRC and QRC-Huber perform similarly is unsurprising as the largest possible reward in any Minatar game is $+1$, a design decision made in part because many algorithms—such as DQN—are unstable when learning from large rewards. Additional results on the Minatar suite are included in Appendix D.

## 7 CONCLUSION

In this work, we extended the saddlepoint reformulation of the mean squared Bellman error, introducing a novel pair of robust losses, the mean absolute Bellman error and the mean Huber Bellman error. We demonstrated that the solutions to these robust objectives are comparable to the MSBE, and in some scenarios are significantly better according to the value error. The resultant gradient-based algorithms are less sensitive to choice of stepsize in prediction and have more stable performance distributions in control.

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

## A    BICONJUGATE FORMS

**Proposition A.1** *The biconjugate of the square function $f(x) = \frac{1}{2}x^2$ is $f^{**}(x) = \max_{h \in \mathbb{R}} hx - \frac{1}{2}h^2$.*

**Proof:**  Recall the definition of the convex conjugate and correspondingly the biconjugate:

$$f^*(x) = \sup_{h \in \mathbb{R}} \{hx - f(h)\}$$

$$f^{**}(x) = \sup_{h \in \mathbb{R}} \{hx - f^*(h)\}.$$

Then the conjugate of the square function with dual parameter $a$,

$$f^*(x) = \sup_{a \in \mathbb{R}} xa - \frac{1}{2}a^2.$$

Applying the convex conjugate again and we obtain

$$f^{**}(x) = \sup_{h \in \mathbb{R}} \left( xh - \sup_{a \in \mathbb{R}} \left( ha - \frac{1}{2}a^2 \right) \right).$$

Clearly, the inner supremum is achieved at $a^* = h$, so plugging in the maximizing value of $a$, we obtain

$$f^{**}(x) = \sup_{h \in \mathbb{R}} \left( xh - h^2 + \frac{1}{2}h^2 \right)$$

$$= \max_{h \in \mathbb{R}} \left( xh - \frac{1}{2}h^2 \right).$$

Finally multiplying by two, $2f(x) = x^2$ and $2f^{**}(x) = \max_{h \in \mathbb{R}} \left( 2xh - h^2 \right)$, arriving at the biconjugate of the square function used in Dai et al. (2017) and Dai et al. (2018). □

**Proposition A.2** *The biconjugate of the absolute value function $f(x) = |x|$ is $f^{**}(x) = \max_{h \in [-1,1]} xh$.*

**Proof:**  The proof follows the same format as the proof of the biconjugate for the square function. Defining the conjugate and biconjugate respectively,

$$f^*(x) = \sup_{a \in \mathbb{R}} xa - |a| = \begin{cases} 0 & \text{when } |x| \leq 1 \\ \infty & \text{otherwise.} \end{cases}$$

$$f^{**}(x) = \sup_{h \in \mathbb{R}} xh - f^*(h).$$

Simplifying the biconjugate form, we get

$$f^{**}(x) = \sup_{h \in \mathbb{R}} \begin{cases} xh & \text{when } |h| \leq 1 \\ xh - \infty & \text{otherwise.} \end{cases}$$

$$= \sup_{|h| \leq 1} xh \quad \triangleright \ -\infty \text{ is not feasible}$$

$$= \sup_{h \in [-1,1]} xh.$$

Finally, considering the maximizing values of $h$, we get $h = -1$ when $x < 0$ and $h = 1$ with $x > 0$ so the biconjugate simplifies to

$$f^{**}(x) = \text{sign}(x)x = |x| = f(x)$$

thus completing the proof. □

**Proposition A.3** *The biconjugate of the huber function is $f_\tau^{**}(x) = \max_{h \in [-\tau, \tau]} xh - \frac{1}{2}h^2$.*

**Proof:** Let the huber function be defined as

$$p_\tau(a) = \begin{cases} \frac{1}{2}a^2 & \text{if } |a| \leq \tau \\ \tau|a| - \frac{1}{2}\tau^2 & \text{otherwise.} \end{cases}$$

Then the convex conjugate is

$$f^*(x) = \sup_{a \in \mathbb{R}} xa - p_\tau(a)$$

$$= \sup_{a \in \mathbb{R}} \begin{cases} xa - \frac{1}{2}a^2 & \text{if } |a| \leq \tau \\ xa - \tau|a| + \frac{1}{2}\tau^2 & \text{otherwise.} \end{cases}$$

Resolving the supremum, consider first the case that $|x| \leq \tau$, then

$$\sup_{|a| \leq \tau} xa - \frac{1}{2}a^2$$

$$= xa^* - \frac{1}{2}a^{*2} \qquad\qquad \rhd \; a^* = x$$

$$= x^2 - \frac{1}{2}x^2$$

$$= \frac{1}{2}x^2$$

for the first case of the piecewise, and for the second case, first consider $0 \leq x \leq \tau$, then

$$\sup_{|a| > \tau} xa - \tau|a| - \frac{1}{2}\tau^2$$

$$= \sup_{a > \tau}(x - \tau)a - \frac{1}{2}\tau^2$$

$$= -\infty$$

because $x - \tau \leq 0$, and when $-\tau \leq x < 0$, we have

$$\sup_{|a| > \tau} xa - \tau|a| - \frac{1}{2}\tau^2$$

$$= \sup_{a < -\tau} xa - \tau|a| - \frac{1}{2}\tau^2$$

$$= -x\tau - \tau^2 - \frac{1}{2}\tau^2 \qquad\qquad \rhd \; a^* = -\tau$$

$$= -x\tau - \frac{3}{2}\tau^2.$$

Finally, because $-\tau \leq x < 0$ we have $-x\tau - \frac{3}{2}\tau^2 \leq \frac{1}{2}x^2$, so the maximum is $\frac{1}{2}x^2$.

Next consider the second case that $|x| > \tau$, then we need not consider all sub-cases because when $x > \tau$, then

$$\sup_{|a| > \tau} xa - \tau|a| - \frac{1}{2}\tau^2$$

$$= \sup_{a > \tau}(x - \tau)a - \frac{1}{2}\tau^2$$

$$= \infty$$

and when $x < -\tau$ then

$$\sup_{|a| > \tau} xa - \tau|a| - \frac{1}{2}\tau^2$$

$$= \sup_{a < -\tau} xa - \tau|a| - \frac{1}{2}\tau^2$$

$$= \infty$$

and the supremum will be $\infty$ when $|x| > \tau$.

Thus the conjugate function is

$$f^*(x) = \begin{cases} \frac{1}{2}x^2 & \text{if } |x| \leq \tau \\ \infty & \text{otherwise.} \end{cases}$$

Using this to compute the biconjugate, we obtain

$$f^{**}(x) = \sup_{h \in \mathbb{R}} hx - f^*(h)$$

$$= \sup_{h \in \mathbb{R}} \begin{cases} hx - \frac{1}{2}h^2 & \text{if } |h| \leq \tau \\ hx - \infty & \text{otherwise} \end{cases}$$

$$= \max_{h \in [-\tau, \tau]} hx - \frac{1}{2}h^2$$

where again the constraints on the maximization come from encoding the infeasibility of $|h| > \tau$. $\square$

### A.1 Proof of Theorem 3.2

We start by showing that the mean absolute Bellman error bounds the mean absolute value error. We then show in Lemma A.5 that the Huber function is a close approximation of the absolute function, and so emits an upper-bound with some approximation error controlled by the Huber parameter $\tau$. Putting these together, then, we show that the mean Huber Bellman error upper bounds the mean absolute *value* error with a small non-vanishing approximation term.

**Lemma A.4** *For any vector $v \in \mathbb{R}^d$, then*

$$\|v_\pi - v\|_1 \leq \|(I - P_{\pi,\gamma})\|_1^{-1} \|\mathcal{T}v - v\|_1.$$

**Proof:** First notice that $v_\pi - (I - P_{\pi,\gamma})^{-1} r_\pi$ where $r_\pi$ is the expected reward function with respect to policy $\pi$. Then we have

$$\begin{aligned} \mathcal{T}v - v &= r_\pi - P_{\pi,\gamma}v - v & \triangleright \text{ by definition of } \mathcal{T}v \\ &= r_\pi - (I - P_{\pi,\gamma})v & \triangleright \text{ rearrange terms to group } v \\ &= (I - P_{\pi,\gamma})v_\pi - (I - P_{\pi,\gamma})v & \triangleright r_\pi = (I - P_{\pi,\gamma})v_\pi \\ &= (I - P_{\pi,\gamma})(v_\pi - v) & \triangleright \text{ rearrange terms} \end{aligned}$$

Bringing the $(I - P_{\pi,\gamma})$ term to the left side, then

$$(I - P_{\pi,\gamma})^{-1}(\mathcal{T}v - v) = v_\pi - v.$$

Finally, because the matrix norm induced by the 1-norm is compatible, we get that

$$\|(I - P_{\pi,\gamma})^{-1}\|_1 \|\mathcal{T}v - v\|_1 \geq \|v_\pi - v\|_1$$

thus completing the proof. $\square$

**Lemma A.5** *Let $\tau_{cap} = \min(\tau, 1)$ and $\tau > 0$, then for any vector $a \in \mathbb{R}^d$ and $0 < \epsilon \leq \tau_{cap}^2$*

$$\|a\|_1 \leq \sum_{i=0}^{d} \frac{\sqrt{\epsilon}}{2\epsilon} p_\tau(a_i) + \frac{\sqrt{\epsilon}}{2}.$$

**Proof:** Our goal is to show $|a| \leq C p_\tau(a)$ for any $a \in \mathbb{R}$. First notice that if $|a| \geq \tau_{\text{cap}}$, then $p_\tau(a) \geq \tau_{\text{cap}}|a|$ by definition of the Huber function and so we are done with this case. However, if $|a| < \tau_{\text{cap}}$, then $p_\tau(a) = a^2$. Should we try to find some constant $C$ such that $|a| \leq Ca^2$, then we easily find that $C \geq \frac{1}{|a|}$ which goes to infinity as $a$ goes to zero. Instead, we can find $|a| \leq C(a^2 + \epsilon)$ for arbitrary $\epsilon > 0$, which yields $C \geq \frac{|a|}{a^2 + \epsilon}$ which is bounded. To find $C$, we have

$$\begin{aligned} C &= \max_{|a| \leq \tau_{\text{cap}}} \frac{|a|}{a^2 + \epsilon} \\ &= \max_{0 \leq a \leq \tau_{\text{cap}}} \frac{a}{a^2 + \epsilon} \\ &= \frac{\sqrt{\epsilon}}{2\epsilon} \end{aligned}$$

because the maximum is obtained at $a = \sqrt{\epsilon}$. We now have that, for $|a| < \tau_{\text{cap}}$ and $C = \frac{\sqrt{\epsilon}}{2\epsilon}$ then $|a| \leq C p_\tau(a)$. Choosing $C \geq 1$ to satisfy the $|a| \geq \tau_{\text{cap}}$ case, and $C = \frac{\sqrt{\epsilon}}{2\epsilon}$ otherwise, we thus obtain our restriction on $\tau_{\text{cap}} = \min(\tau, 1)$ completing the proof. $\square$

## B  PROJECTED BELLMAN ERRORS

In Section 3, we described a conjugate form of the Bellman error. This conjugate Bellman error depends on finding the maximizing function $h : \mathcal{S} \to \mathbb{R}$ from the set of all functions $h \in \mathcal{F}_{\text{all}}$. As is highlighted in Sutton and Barto (2018, Chapter 8), the MSBE is not identifiable; an issue inherited by the conjugate form of the Bellman error. This identifiability issue stems from the fact that the mean Bellman error is defined with respect to a non-observable quantity, the states. When states are aliased together due to partial observability, then the agent can see only part of the statespace while optimizing $\theta$ while seeing the entire statespace for optimizing $h \in \mathcal{F}_{\text{all}}$. Clearly this is not a realistic setting. Modifying modifying this example such that the optimization procedure for $h$ has the same partial observability leads towards an identifiable form of the conjugate Bellman error

$$\text{Identifiable BE}(\theta) \overset{\text{def}}{=} \max_{h \in \mathcal{F}_h} \mathbb{E}\left[2\mathbb{E}\left[\delta(\theta) \mid S\right] h(S) - h(S)^2\right]$$

where $\mathcal{F}_h \overset{\text{def}}{=} \{h = f \circ x\}$ such that $h : \mathcal{S} \to \mathbb{R}$ are all functions defined with respect to features $x : \mathcal{S} \to \mathbb{R}^d$. Because $\mathcal{F}_h \subseteq \mathcal{F}_{\text{all}}$, then the solution to the Identifiable BE may be different from the solution to the conjugate BE due to a more constrained optimization on $h$.

As shown in Section 3, the optimal function $h^*(s) = \mathbb{E}[\delta(S) \mid S = s]$. In the finite state setting, we can represent this function as a vector $u \in \mathbb{R}^{|\mathcal{S}|}$ composed of entries $\mathbb{E}[\delta(S) \mid S = s]$; thus the vector $u = \mathcal{T}\hat{v} - \hat{v}$. We can define a projection operator on $u$ as

$$\Pi_{\mathcal{F}_h, d} u \overset{\text{def}}{=} \arg\min_{h \in \mathcal{F}_h} \|u - h\|_d$$

where $d : \mathcal{S} \to [0, 1]$ is a weighting over states. Then assuming that $\mathcal{F}_h$ is a convex set, we can decompose $u = \Pi_{\mathcal{F}_h, d} u + \tilde{u} = h + \tilde{u}$ where $\tilde{u}$ is the component in $u$ that is orthogonal to $h = \Pi_{\mathcal{F}_h, d} u$ in the weighted space $h^\top D\tilde{u} = 0$ for $D \overset{\text{def}}{=} \text{diag}(d)$. From this projection operator, we can define the mean squared Projected Bellman Error

$$
\begin{aligned}
\overline{\text{PBE}} &\overset{\text{def}}{=} \max_{h \in \mathcal{F}_h} \sum_{s \in \mathcal{S}} d(s)(2\mathbb{E}[\delta \mid s] h(s) - h(s)^2) \\
&= \max_{h \in \mathcal{F}_h} \sum_{s \in \mathcal{S}} d(s)(2u(s)h(s) - h(s)^2) \\
&= \sum_{s \in \mathcal{S}} d(s)(2u(s)h(s) - h(s)^2) \quad \triangleright \ h = \Pi_{\mathcal{F}_h, d} u \\
&= \sum_{s \in \mathcal{S}} d(s)(2(h(s) + \tilde{u}(s))h(s) - h(s)^2) \\
&= \sum_{s \in \mathcal{S}} d(s)(2h(s)^2 - h(s)^2) + 2\sum_{s \in \mathcal{S}} d(s)\tilde{u}(s)h(s) \\
&= \sum_{s \in \mathcal{S}} d(s)h(s)^2 + 2\sum_{s \in \mathcal{S}} d(s)\tilde{u}(s)h(s) \\
&= \sum_{s \in \mathcal{S}} d(s)h(s)^2 \\
&= \|\Pi_{\mathcal{F}_h, d}(\mathcal{T}\hat{v} - \hat{v})\|_d^2
\end{aligned}
$$

where the second to last step is because $h$ is orthogonal to $\tilde{u}$ under weighting $d$.

This connection to projected Bellman errors provides some insight for the role of approximating the function $h$, as well as for the robust objectives discussed in Section 3. Each choice of function space $\mathcal{F}_h$ results in a different linear projection on the vector $u$ describing the Bellman error in every state, with each projection producing its own orthogonal component $\tilde{u}$ which is ignored in the resulting projected objective function. For instance, consider a low-rank projection operator $\Pi_{\mathcal{F}_h, d}$.

The resulting projected Bellman error could project potentially high-error states to zero error, thus allowing no function approximation resources to be used to represent the value function in that state. When no projection is used, equivalently when $\Pi_{\mathcal{F}_h, d} = I_{|\mathcal{S}|}$ the trivial projection, then no errors are projected and the values are learned to directly minimize the Bellman error. And when the space $\mathcal{F}_h = \mathcal{F}$ with $h \in \mathcal{F}_h$ and $\hat{v} \in \mathcal{F}$, then we recover the original mean squared projected Bellman error of Sutton et al. (2009).

## C  BIAS OF TDC GRADIENT

In this section, we discuss the biased gradient estimate used by gradient-correction methods such as TDC or QRC-Huber. We show that in the case of linear function approximation, when $h$ is the best linear approximation of $\mathbb{E}[\delta \mid s]$, then the gradient is unbiased for the MSBE. However, this is no longer the case when considering the robust objectives nor is it the case when $h$ must be approximated online. As has been seen in past results (White and White, 2016; Ghiassian et al., 2018; 2020), this biased gradient does not seem to harm TDC's performance—and in fact the lower variance gradient estimate seems to improve empirical performance—however, our own experiments in Section 5 suggest that for the MABE the biased gradient estimates can often prevent the gradient-correction algorithm from learning.

Manipulating the gradient of the conjugate Bellman error, we get

$$
\begin{aligned}
- \nabla_\theta \delta(\theta) h(s) &= h(s) \nabla_\theta v(s) - \gamma h(s) \nabla_\theta v(S') \\
&= \delta(\theta) \nabla_\theta v(s) + \underbrace{(h(s) - \delta(\theta)) \nabla_\theta v(s)}_{\text{extra term}} - \gamma h(s) \nabla_\theta v(S').
\end{aligned}
$$

Let $h^* = \mathbb{E}\left[xx^\top\right]^{-1} \mathbb{E}[x\delta]$ be the optimal linear regression solution for $h$ and let both $h$ and $v$ be parameterized with linear function approximation. Then because $\nabla_\theta v(s) = x(s)$, we have

$$
\begin{aligned}
\mathbb{E}[(h^* - \delta) \nabla_\theta(s) \mid s] \\
= \mathbb{E}\left[x(x^\top h^* - \delta) \mid s\right] \\
= x(x^\top h^* - \mathbb{E}[\delta \mid s]) \\
= xx^\top h^* - x\mathbb{E}[\delta \mid s]
\end{aligned}
$$

and in expectation across all states

$$
\begin{aligned}
\mathbb{E}[(h(S) - \delta) \nabla_\theta(S)] \\
= \mathbb{E}\left[xx^\top\right] h^* - \mathbb{E}[x\delta] \\
= \mathbb{E}\left[xx^\top\right] \mathbb{E}\left[xx^\top\right]^{-1} \mathbb{E}[x\delta] - \mathbb{E}[x\delta] \\
= \mathbb{E}[x\delta] - \mathbb{E}[x\delta] = 0
\end{aligned}
$$

where $x = x(s)$.

Unfortunately, consider the case of the robust objectives. Instead of the $(h(s) - \delta)$ term, we apply a nonlinear transformation to only $h(s)$. Intuitively, in the case of the MABE the difference between $\text{sign}(h(s)) - \delta$ can be arbitrarily large. In the case of the MHBE, the bias due to ignoring this additional term is a function of the clipping parameter $\tau$. Clearly as $\tau \to \infty$, then $\text{clip}_\tau(h(s)) \to h(s)$ and the same argument applies as in the MSBE case.

The bias of the gradient estimate used by gradient-correction methods for the MHBE in the case of linear function approximation is

$$
\begin{aligned}
\mathbb{E}[\|(\text{clip}_\tau(h^*(s)) - \delta(\theta))x\|_\infty] \\
= \mathbb{E}[|\text{clip}_\tau(h^*(s)) - \delta(\theta)|\|x\|_\infty] \\
\leq \sum_{s \in \mathcal{S}} d(s) \begin{cases} 0 & \text{when } |h^*(s)| \leq \tau \\ |\tau \text{sign}(h^*(s)) - \delta(\theta)|\|x\|_\infty & \text{otherwise} \end{cases} \\
= \sum_{|h^*(s)| > \tau} d(s) |\tau \text{sign}(h^*(s)) - \delta(\theta)|\|x\|_\infty.
\end{aligned}
$$

Because when $|h^*(s)| \leq \tau$, then $|\text{clip}_\tau(h^*(s))| = |h^*(s)|$ and we are again in the case of the gradient of the MSBE. However, when $|h^*(s)| \geq \tau$, then $|\text{clip}_\tau(h^*(s))| = \tau$ and we accumulate some bias based on how far $\delta$ is from $\tau$. When $\mathbb{E}[\delta] = 0$, then likewise $h^*(s) = 0$ because the zero vector is always representable by a linear function approximator (by definition of linearity). Because $\tau > 0$, then $\text{clip}_\tau(h^*(s)) = 0$ and the bias is zero, so the fixed point of the algorithm remains unchanged.

# D  ADDITIONAL RESULTS

In this section we include supplementary results to the main body of the paper. We first investigate the relative ordering of the proposed optimization algorithms when measuring the MAVE instead of the MSVE. We then motivate empirically why we built our nonlinear control algorithm based on a gradient-correction method instead of a saddlepoint method by investigating the performance of nonlinear GQ on our benchmark control domains. Finally, we end with two ablation studies investigating the impact of the Huber threshold parameter on each domain as well as a second ablation investigating the impact of the choice to exclude target networks in the main body of the paper, especially for DQN.

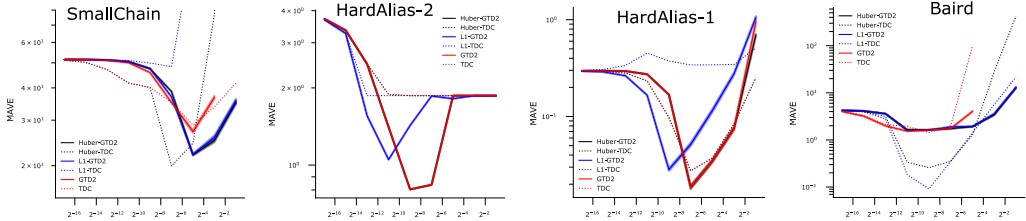

Figure 6: MAVE averaged over 100 independent trials, for each swept stepsize in key prediction domains. The mean squared algorithms generally performed well across environments—even the adversarially chosen environments—suggesting the difficulty in minimizing the MABE. The Huber algorithms performed best across many environments, often displaying less sensitivity to the choice of stepsize.

In Figure 6, we investigate the stepsize sensitivity of each algorithm on four representative linear prediction problems. The robust algorithms generally have similar best performance as the MSBE algorithms, but with less sensitivity to choice of stepsize. The $\ell_1$-TDC algorithm often fails to learn a meaningful value function estimate in the given number of training steps. We hypothesize that this is due to the bias in the gradient estimate for gradient-correction methods, which is pronounced in the case of the MABE but not in the case of the other objectives. The results in Figure 6 are averaged over 100 independent trials for every choice of stepsize, algorithm, and domain. The shaded regions correspond to standard errors, though error bars are excluded from the TDC algorithms for readability and because the standard errors are negligible. The performance measure for each algorithm is the average error over the last 25% of steps in each domain.

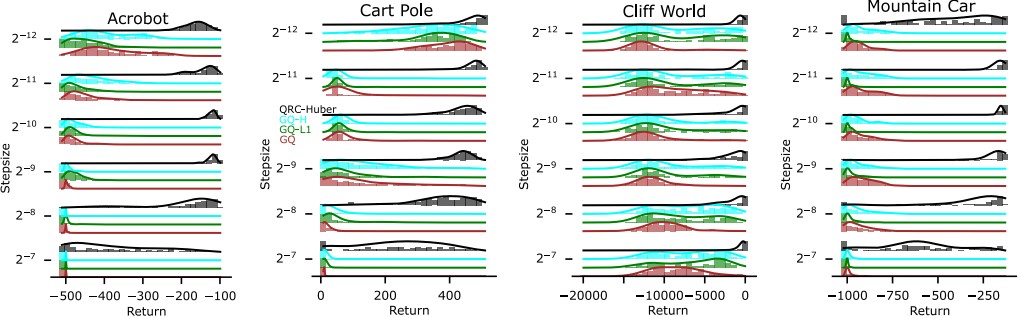

Figure 7: Comparing the mean return over the last 25% of steps across several saddlepoint methods against QRC-Huber. The saddlepoint methods generally perform very poorly, frequently finding a policy only slightly better than the random policy. These results are consistent with the findings of Ghiassian et al. (2020) and motivate building on gradient-correction methods for nonlinear control. Like QRC-Huber, GQ-Huber uses a twice differentiable estimate of the clip function and all algorithms use the ADAM optimizer.

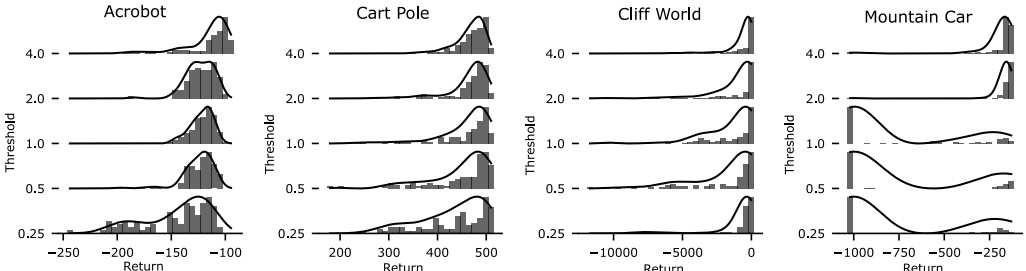

Figure 8: Ablating the impact of the threshold parameter for the Huber loss function for the QRC-Huber algorithm across the benchmark domains. For three of the domains, QRC-Huber is robust to the choice of threshold parameter with a default value of $\tau = 1$ being a good choice. However, the Mountain Car domain shows high-bimodality in performance distribution across multiple random initializations of the neural network for smaller values of the threshold parameter.

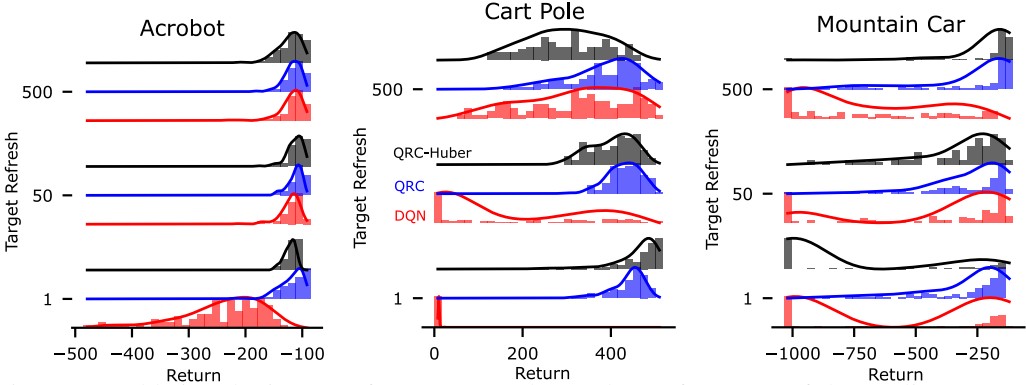

Figure 9: Ablating the impact of target networks on the performance of the nonlinear control algorithms on benchmark domains. The gradient-based methods receive much less benefit from using target networks than DQN, which requires target networks to achieve above random performance on Cart-pole and to reduce the bimodality of its performance on Mountain Car. Even with target networks, DQN still exhibits large skew and bimodality in its performance distributions, indicating instability.

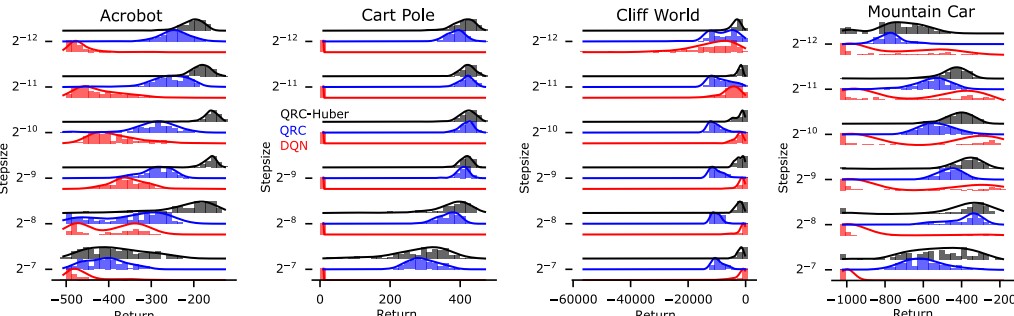

Figure 10: Comparing algorithms on benchmark control domains with the area under the learning curve as the performance metric. Unlike Figure 4, early learning is included in the performance metric, giving a sense of the sample complexity of each algorithm. QRC-Huber tends to perform favorably across all four domains compared to QRC and DQN, exhibiting much more narrow performance distributions that are often centered around higher rewards than the competitor algorithms.

Figure 7 investigates the saddlepoint optimization algorithm for nonlinear control across our four benchmark domains. Generally, the Greedy-GQ algorithm performs considerably worse than gradient-correction algorithms; a motivating factor for building on gradient-correction methods (Dai et al., 2018; Ghiassian et al., 2020). A possible explanation for this poor performance is the dependency of the representation learning process on having an accurate estimate of $h(s)$, which itself depends on having a well-learned representation. This circular dependency is less obviously present in gradient-correction methods, which depend on a sample of the error signal instead of an estimate of the error

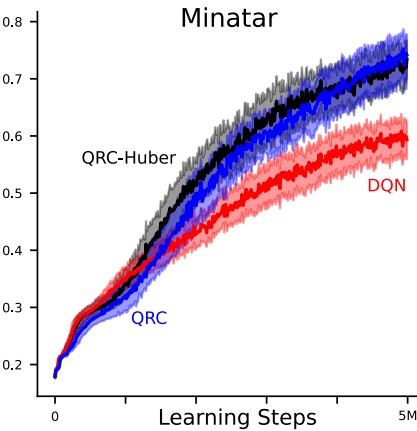

Figure 11: Evaluating the performance of each control algorithm on the Minatar suite of games. The learning curves show the scaled performance metric averaged across domains with 95% bootstrapped confidence intervals about the mean. Because each point in the learning curve has less underlying structure than the aggregate performance metric, the confidence intervals are significantly more pessimistic than reported in Table 1. As such, the sample mean performance of QRC-Huber is slightly higher than QRC during early learning, but not statistically significantly so in this result. Both gradient-based algorithms considerably outperform DQN with statistical significance, indicating both less domain-sensitivity to meta-parameters as well as better absolute performance.

signal for the primary learning process. Improving the performance of saddlepoint optimization methods for nonlinear control is important future work.

### D.1 ABLATING DESIGN DECISIONS

One of the proposed robust objectives depends on a new meta-parameter—$\tau$ the threshold for the Huber function—which was not present in previous extensions of the conjugate Bellman error to nonlinear control. While for most of our domains we could reasonably pick a default value of $\tau = 1$ and avoid allowing our proposed algorithm more opportunities to tune meta-parameters, this choice impacted the claims made in the main body of the paper. The choice of $\tau$ depends on the magnitude of the TD errors experienced by the algorithm during optimization, which is driven in-part by the magnitude of the rewards, and thus is domain-dependent. To decrease this dependency, we could consider scaling the magnitude of rewards in a domain-independent way, for instance using PopArt (Hessel et al., 2018).

In Figure 8, we ablate over several choices of threshold parameter for the QRC-Huber algorithm. We likewise investigated the sensitivity of DQN to choice of threshold parameter, but due to the general instability of DQN it was less clear which values of $\tau$ were generally best. Allowing DQN to select its best $\tau$ for each domain does not change the conclusions in Figure 4, so for simplicity we choose to maintain consistency between DQN and QRC-Huber. We see long-tail performance distributions as the threshold parameter is made smaller, likely due to the optimization process spending more time in the mean absolute region of the Huber loss. On the Mountain Car domain, both QRC-Huber and DQN were significantly impacted by $\tau < 2$ and saw strongly bimodal performance distributions.

The poor performance of DQN in Figure 4—especially on the Cart-pole domain—is surprising; however recent work has also shown that DQN has shockingly poor performance on a wide variety of domains (Obando-Ceron and Castro, 2021). A potential explanatory factor for the poor performance could be the choice to exclude target networks from our investigation, which could adversely affect DQN disproportionately compared to the gradient-based algorithms. To understand the impact of our choice to not use target networks, we ablate over the number of steps taken between synchronizing the weights of the target network.

Figure 9 shows the number of steps between target network synchronization for QRC-Huber, QRC, and DQN, where one step of synchronization refers to not using target networks at all. For Acrobot and Cartpole, increasing the number of steps between synchronization appears to harm the performance of the gradient-based methods, likely due to artificially reducing the speed that the bootstrapping targets receive new information. DQN, on the other hand, benefits from the reduced variance in the bootstrapping targets and tends to perform better across all three domains as the target networks are updated less frequently; though even with 500 steps between synchronization, DQN performs poorly on Mountain Car.

## E  EXPERIMENTAL DETAILS

### E.1  ENVIRONMENTS

In this section we provide further details for the environments and problem settings used in Sections 5 and 6.

**HardAlias-1** is an 8-state random walk where the agent starts in the far left state and moves right with 90% probability. The episode terminates on taking the move right action in the far right state. The agent receives $-1$ reward per step with a discount factor of $\gamma = 0.99$. The first, third, and final states all share a common feature, and the remaining five states use the dependent feature representation from Sutton et al. (2009), resulting in a feature vector of size $d = 4$ for each state.

The second hard aliasing problem, **HardAlias-2**, is the 2-state problem from Tsitsiklis and Van Roy (1997). We lightly modify the reward function of the MDP so the optimal value function cannot be perfectly represented, thus allowing each objective to have different minima. The agent receives a reward of $+1$ after transitioning from the first state to the second, and a reward of 0 for all other transitions.

**Outlier** is a large 49-state random walk with an additional "entry" state, where the agent has an $\epsilon = 0.01$ chance of terminating immediately with -1000, or a $1 - \epsilon$ chance of entering the middle state of the random walk. To emulate a more realistic learning scenario, we use a randomly initialized frozen neural network with ReLU activations and two hidden layers of sizes ten and five respectively to generate five features to describe 50 states. Taking the left action in the far left state of the random walk results in termination and a reward of $-1$ and correspondingly the right action in the right state results in termination with a reward of $+1$. The discount factor is set to $\gamma = 0.99$ and the left action was chosen with probability $\epsilon$ in every state.

For the two random walks, the first with $N = 5$ states (**SmallChain**) and the other with $N = 19$ states (**BigChain**), we use a randomly initialized neural network representation with ReLU activations and three hidden layers of sizes $h_1 = 4N$, $h_2 = N$, and $h_3 = \frac{N}{2}$ units respectively, resulting in a feature representation of size $d = h_3$. These problems are off-policy with the target policy taking the left action with 90% probability and the behavior policy taking both actions with equal probability. The discount is $\gamma = 0.99$ for both problems.

Finally, **Baird** is the well-known star MDP from Baird (1999). It is used to investigate optimization performance in a high-variance off-policy setting where TD diverges. We do not use it to evaluate the quality of fixed points, because the linear function approximation can represent the true values, and so the fixed-points of each objective are equal in quality.

In the nonlinear experiments, for all domains we use a discount factor of $\gamma = 0.99$ and an $\epsilon$-greedy policy with $\epsilon = 0.1$. In Mountain Car, Acrobot, and Cliff World, the agent receives a reward of -1 per step until termination and in Cart-pole the agent receives a reward of +1 per step. Every environment has an episode cutoff if the agent fails to reach a terminal state with a pre-specified number of steps. When cut off, the agent is teleported back to the start state and does not update its value function, thus preventing the agent from bootstrapping over the teleportation transition. All algorithms are run for one hundred thousand steps in total across all episodes, except in Lunar Lander where algorithms are run for 250k steps.

In the Mountain Car environment (Moore, 1990; Sutton, 1996), the goal is to drive an underpowered car to the top of a hill. The agent receives as state the position and velocity of the car, and can choose to accelerate forward, backward, or to do nothing on each timestep. The episode terminates when the agent reaches the top of the hill, or is cut off when the agent reaches a maximum 1000 steps. In the Cart-pole domain (Barto et al., 1983), the agent balances a pole attached to a cart which can move along a single axis. The agent receives as state the position and velocity of the cart, as well as the angle and angular velocity of the pole. The episode ends when the pole falls or if the agent reaches the maximum of 500 steps. Finally, the Acrobot domain (Sutton, 1996) has the agent swing a double-jointed arm above a threshold by moving only the inner joint. The agent receives as state the current angle and velocity of the joints and can take as action, swing left or swing right. The episode terminates when the specified height is achieved, or is cut off after 500 steps.

The Cliff World environment—lightly adapted from Sutton and Barto (2018)—is a discrete gridworld with 20 states. The agent starts in the bottom left state and seeks to reach the goal state in the bottom

right. Along the bottom of the grid lies a cliff, where the agent receives a large penalty of -1000 reward for stepping into the pit and is teleported back to the initial state *without* terminating the episode. The episode terminates only when the agent reaches the goal state, or is cut off when the agent reaches a maximum of 500 steps.

For all environments, we fix meta-parameters other than the stepsize to their default values. For QRC-Huber, we fix regularizer parameter $\beta = 1$ and secondary stepsize ratio $\eta = 1$. For both mean Huber algorithms, we fix the Huber threshold parameter $\tau = 1$ for all domains except Mountain Car, where we use $\tau = 2$. We further ablate the impact of this decision in Section D. We sweep over the stepsize parameter for all algorithms and environments and report results for every swept stepsize.

### E.2   FINDING FIXED-POINTS

To find the fixed-points of the objectives in Section 5, we used an iterative optimization procedure that assumed access to the underlying dynamics of each MDP to compute exact expected gradients for each update to the primary variable. We use first order stopping conditions to ensure that the optimization procedure has reached a fixed-point; i.e. when the norm of the gradient is near zero (specifically less than $10^{-7}$). We use ADAM parameters of $\beta_1 = 0.99$ and $\beta_2 = 0.999$ along with a moving iterate average with exponential moving average parameter $\beta = 0.9$ to reduce oscillation of the gradients and iterates around the fixed-point (especially for the mean absolute objective, where we performed subgradient descent). We decay the global stepsize according to $\alpha = \frac{1}{\sqrt{t}}$ where $t$ is the number of update steps taken so far. [2]

### E.3   LINEAR PREDICTION

For the prediction problems comparing optimization methods in Section 5, we swept over the primary and secondary stepsize for all algorithms allowing each to be tuned independently. We swept values of the primary stepsize $\alpha \in \{2^{-1}, 2^{-2}, \ldots, 2^{-10}\}$ and the ratio between the primary and secondary stepsize $\eta = \frac{\alpha_\theta}{\alpha_h} \in \{2^{-6}, 2^{-4}, \ldots, 2^0, \ldots, 2^6\}$. All algorithms have the same number of meta-parameter combinations, so comparison between each algorithm remains fair. Reported results use SGD with a constant stepsize, though results using RMSProp yielded similar conclusions and thus were omitted. All algorithms were evaluated after 10k updates for each domain except the random walk which required 100k updates to reasonably converge.

### E.4   NONLINEAR CONTROL

For all of the nonlinear control algorithms and domains, we used neural network function approximation with two hidden layers and ReLU activation units. For Acrobot, Mountain Car, and Cliff World we used 32 hidden units in both layers, and in Cart Pole we noticed significantly better performance for all algorithms when using 64 hidden units (consistent with the findings of Obando-Ceron and Castro (2021) which suggested Cart Pole needs considerably more parameters for good performance), finally for Lunar Lander we used 128 hidden units in both layers. We use experience replay buffers to store the 4000 (10k for Lunar Lander) most recent transitions and draw 32 samples to compute mini-batch updates on every timestep. We use the ADAM optimizer with default parameters for all algorithms ($\beta_1 = 0.9$ and $\beta_2 = 0.999$), but notice little difference in conclusions when using SGD or RMSProp optimizers. All agents are trained using $\epsilon$-greedy behavior policies, with $\epsilon = 0.1$ for every domain. Agents are trained for a fixed 100k steps for Acrobot, Mountain Car, and Cart Pole. Cliff World only required 50k steps to achieve good policies, and Lunar Lander required 250k steps.

For the Minatar games, we used the same function approximation architecture and meta-parameter settings as in Young and Tian (2019). Specifically the neural network uses a single convolutional layer with 16 channels, a stride-width of 1, and a kernel-width of 3 followed by a ReLU activation, the output of the convolutional layer is then flattened and sent to a single fully-connected layer with 128 hidden units and ReLU activation. We use the ADAM optimizer (Kingma and Ba, 2015) with default parameters and sweep over stepsizes in $\alpha \in \{2^{-13}, 2^{-12}, \ldots, 2^{-8}\}$. For DQN only, we additionally

---

[2]We note that the MSBE fixed-point can easily be computed analytically using a least-squares solver. For consistency, we use the iterative solver even for the MSBE. Reported results and conclusions are unchanged when using the analytical solutions.

sweep over target network refresh rates in $\{1, 8, 32, 64\}$ steps. Experiments are run for 5M steps for each domain and a replay buffer of size 100k is used.

### E.5 MINATAR EXPERIMENTAL PROCEDURE

For the Minatar demonstration, we treat the Minatar domain suite as a single problem setting. In doing so, we can take advantage of lower variance performance metrics by averaging performance over each of the domains, allowing us to report statistically significant claims using far fewer computational resources. The procedure is as follows.

1. We first swept over several choices of stepsize using only five runs for each game.
2. We then scaled the AUC for each individual run using probabilistic performance profiles (Barreto et al., 2010) to a value between $[0, 1]$.
3. We picked the best performing stepsize for each algorithm by averaging the scaled AUC across runs and across games.
4. Finally we ran an additional 30 runs for each algorithm on each game using that algorithm's best stepsize for a total of 150 runs and report the average scaled performance in Table 1 as well as the average scaled performance over time in Figure 11.

### E.6 COMPUTATIONAL RESOURCES

For this paper, we used approximately eight CPU years of compute on a general purpose CPU cluster with modern hardware. We did not use GPUs for any experiment, nor other specialized hardware for training our models. We used the Torch library (Paszke et al., 2019) for defining neural networks and autodifferentiation for the nonlinear control experiments, and used the numpy library (Harris et al., 2020) for the linear prediction experiments.

