# OpenReview forum: "Robust Losses for Learning Value Functions"
_ICLR.cc/2022/Conference — ICLR 2022 Submitted_

### Official Review · Reviewer_78BK · 2021-11-01

**Correctness:** 4
**Technical Novelty And Significance:** 2
**Empirical Novelty And Significance:** 2
**Recommendation:** 5
**Confidence:** 2

**Main Review:**

The robustness is important in RL learning. The MHBE defined in this paper is inituitive. And the authors also develop reformulation to solve it in practive.

**Summary Of The Paper:**

The paper proposes a mean Huber value error in the TD-learning. And the paper demonstrates the robustness under such loss.

**Summary Of The Review:**

The major contribution is to develop a new type of loss, which is not sufficient according to the ICLR standards.

My main concern is that the contribution is not sufficient. I think the main contribution of this paper is the introduction of a robust loss. The conjugate reformulation is standard. And the bound developed in Theorem 3.2 is not surprising.

The writing is also not clear. For example, MHBE and MABE are never defined.

---

> ### Author Response · Authors · 2021-11-10
> **Author response**
>
> We solve an open problem and provide a thorough empirical and theoretically sound investigation. Using robust variants of the BE has been an open problem. In fact, we have even tried to solve it in the past and did not know how to do so. The introduction of the conjugate form of the BE helps us find a path. It is simple in hindsight, but that does not make it trivial. We think the following two things are exactly what ICLR is looking for: (a) solving an open problem and (b) providing a careful investigation into the properties of the robust losses and different algorithms to optimize it.
>
> If you could clarify why it is below the bar, then we can address those comments.
>
> The MHBE and MABE are both defined throughout the paper. The reviewer references this definition in their “Main Review” section, stating the definition is “intuitive”. A few concrete examples:
> * Textually in paragraph 6, sentence 2.
> * Mathematically at the bottom of page 4.
> * And relevant update rules in equations (3) and (4).

---

> > ### Comment · Reviewer_78BK · 2021-11-14
> > **Novelty**
> >
> > My concern is still on the novelty. What are the fundamental difficulties on the extension of the conjugate method introduced in [Dai et al. 2017,2018] to MHBE and MABE?
> >
> > And what are the methodology contributions of this paper compared to [Dai et al. 2017,2018] besides the introduction of new types of errors?

---

> > > ### Author Response · Authors · 2021-11-17
> > > **Response**
> > >
> > > It is not fundamentally difficult to extend the conjugate method from Dai et al. to the MHBE. Our contribution is the classic one of recognizing that we can put two things together to solve a problem, then smoothing out the rough patches (e.g. deriving a conjugate form of the Huber function and proposing a constrained optimization approach). The difficulty of the solution (or lack of difficulty) does not preclude the key novelty, that is we solve the open problem of using robust versions of the BE. This had not been solved. The primary contribution is to provide this simple solution and then---especially---to investigate it carefully.
> > >
> > > We would further like to add that, though it is obvious now in retrospect, using this conjugate form to reformulate the BE for these other losses is not at all obvious from Dai et al. Once we understood that paper well, we were able to pursue this direction. If someone in RL wanted to suddenly use the MHBE, instead of the MSBE, they would not just obviously be able to say: "oh, I know, I'll just use the conjugate form." The role of this paper is that we take an open problem and provide a solution that is not obvious to most people in RL.
> > > Beyond the conjugate form, this work is quite different from Dai et al and makes several methodology contributions:
> > >
> > > * We formalize the relationship between the saddlepoint algorithms (i.e. GTD2-like algorithms such as Dai et al.'s analysis) and what we call "gradient correction" algorithms (i.e. TDC such as TDRC and QRC from (Ghiassian et al. 2020)). Building on this relationship, we propose a novel control algorithm that is fundamentally different from Dai et al.
> > > * We study properties of the MDPs that lead to favorable fixed-points for each of the various BE objectives, relating these objectives to the on-going discussion about learnability/identifiability from e.g. Baird 1995, Sutton et al. 2009, Scherrer 2010, and Sutton et al. 2018. This property of the objectives is neither discussed or empirically investigated in Dai et al.
> > > * A primary motivation of these gradient-based objectives is improved learning stability and guaranteed convergence. Neither of these properties are measured directly in Dai et al. empirically. Our paper novelly measures and investigates both properties directly, providing novel empirical insight to the various distributions of performance found by semi-gradient methods (i.e. long-tailed distributions with some runs failing catastrophically, bimodal distributions where some percent of runs perform excellently and the rest perform very poorly, etc.), and showing that gradient-based methods such as QRC-Huber do not generally suffer the same instability on our investigated problem settings; even without using target networks.

---

### Official Review · Reviewer_VsHs · 2021-11-01

**Correctness:** 4
**Technical Novelty And Significance:** 3
**Empirical Novelty And Significance:** 4
**Recommendation:** 8
**Confidence:** 4

**Main Review:**

Strengths:
* A nicely executed paper that starts from a simple premise and connects several prior ideas together -- namely the TDC/GTD2 algorithms with the conjugate formulation of Bellman error minimization and uses this to derive a more robust versions of the aforementioned algorithms.
* Empirical results investigate both a simpler setup with linear prediction experiments as well as a more end to end non-linear control experiment to validate the practical benefits for policy optimization.
* One of the novel conceptual contributions in this paper, appears to be making a connection between the auxiliary model $h$ used in the conjugate formulations of the Bellman error with the very differently motivated "secondary weights" model in GTD2 and TDC algorithms, where the additional model $h$ in both cases can be viewed as attempting to predict a residual error corresponding to the "primary weights" of the value function.


Request for clarifications / other comments for improvement:
* For the empirical results on non-linear control problems (Figures 4, 5 and Table 1), it is somewhat unclear how much of a role is played by the auxiliary variable learning mechanism versus simply changing the loss from squared error to huber loss. For example, it would be interesting to see how a more naive baseline like semi-gradient update corresponding to the DQN + with Huber loss metric instead of DQN + squared error performs alongside the other three curves in each of the environments.
* The main algorithmic proposal is based on the closely related prior works TDC, GTD2 and QRC, yet this aspect is somewhat buried until fairly late in the paper.
* For the last term in Theorem 3.2, $p_\tau$ is applied to the vector $\mathcal{T} v - v$, but without specifying any reduction operation -- looking at the proof, it seems to me that there might be a need to add a max operation over the states to this bound. Could you please confirm? The combination of the multiplicative term involving a matrix inverse norm and the max operation over the states above makes this a potentially fairly weak bound in practice.
* Another case of using an auxiliary model that learns to predict the TD errors has also been proposed for improving learning in actor-critic models, e.g. "Characterizing the gap between AC and Policy gradient", Wen et. al., ICML 2021.


Minor comments
* In Equation (4), the notation for state seems inconsistent between t and t+1 across $s, S'$.
* In the equation for QRC-Huber updates for the auxiliary variable, there is some inconsistent notation across the references $\theta_h, \theta_{ht}, \theta_{h, t+1}$. Also, the variable $\beta$ in the secondary variable update needs a definition or reference of some sort.


**Summary Of The Paper:**

This paper starts with the premise that squared error minimization, despite its wide use, might not be the most effective option for learning value functions. The authors hypothesize that this might be because of squared error's emphasis on outlier states where the bellman error is large at the expense of accuracy on other states. To address this, they consider the absolute value and huber errors alongside the squared error objective and propose a saddle-point reformulation for these objectives that requires learning an auxiliary learned state function which, essentially, attempts to predict the residual error at each state.

Based on this motivation, starting with Section 4 the authors then make a connection of the proposed robust loss framework involving a learned auxiliary state function with prior work on algorithms for improving TD learning, namely, GTD2 and TDC.  Empirical analysis covers both prediction tasks as well as control tasks. In the prediction experiments, evaluations are conducted over carefully designed synthetic problems with linear function approximation in order to highlight particular challenges for each objective. These demonstrate that the Huber objective can often improve the prediction accuracy over squared error over a wide range of step sizes (learning rates).

For the empirical investigation on non-linear control problems, the authors consider a Huber-extension of the previously proposed QRC algorithm (which itself extends the TDC update to be considered in conjunction with the DQN target definition) and demonstrate that QRC-huber can improve on baselines in certain environments while being competitive elsewhere. These algorithms are also validated on a mini version of the Atari domain called Minatar.

**Summary Of The Review:**

This is a nice paper that makes a novel connection between the secondary variable update in GTD2/TDC with the conjugate formulation of Bellman errors involving an auxiliary state function (both of which involve predicting the residual error with a separate model). While the technical contributions in Section 3 (e.g. Theorem 3.2) are not particularly significant, I believe the main value in the paper is the conceptual linking of two different lines of work to derive an improved algorithm over well motivated baselines. The empirical evaluation is well motivated and quite thorough, even if only for a limited set of benchmarks.

---

> ### Author Response · Authors · 2021-11-10
> **Author response**
>
> **First, we very much appreciate this reviewer’s quality and detailed review.**
>
> ### DQN with Huber loss
> Actually, the DQN algorithm used throughout the empirical investigation does use the Huber loss instead of a mean squared loss, consistent with the original DQN paper. However, we recognize that the name “DQN” has been taken to mean many variations of the same algorithm and so we will clarify in Section 6 exactly what components of DQN we are using in our experiments. We additionally performed an ablation on DQN’s Huber threshold parameter, however, the results did not make the cut for this draft. On several domains, the use of a larger threshold (more similar to a mean squared loss) improved the performance of DQN marginally, however, the stability of this mean squared DQN remained quite poor. This mean squared DQN likewise performed very poorly on the CliffWorld counterexample domain, similar to the other mean squared algorithm: QRC.
>
> ---
>
> ### Relation to prior works
> We see the reviewer’s point. We mention GTD2 and TDC early in the submission (paragraph 2) as a primary motivation for our work but will extend this discussion to highlight that our work specifically is building off of these prior works. We will likewise include the QRC citation in the discussion in paragraph 3, where we discuss our relationship to other (bi)conjugate optimization methods like SBEED (which was a major motivating algorithm for the QRC paper).
>
> We are familiar with Wen et al., 2021 and believe it is only marginally relevant and so chose not to cite it due to space limitations. There is a large class of papers that utilize estimates of the TD error to improve optimization properties (advantage estimation belongs to this class, hence the connection to actor-critic methods), however, these works often build on bias-variance claims, while our work focuses more specifically on fixed-points of varying objectives. With the additional space allowed in the camera-ready, we will include a brief discussion of this literature and its relation to the optimization properties of our proposed algorithms.
>
> ---
>
> ### Bound proof
> The Huber function is applied element-wise to Tv - v, with the reduction operation (a mean) used after this application. We agree that this is a slight abuse of notation (we previously defined the Huber function as a scalar-valued function) and so have clarified this further in the proof.
>
> ---
>
> ### Minor comments
> * Thanks for pointing out the inconsistency with S, S’ and using time indices. We will fix these!
> * Likewise, we will add the time index to $\theta_h$ for these equations. We note that the reference for $\beta$ was provided above the update equation (it is from QRC i.e. [Ghiassian et al., 2020]). However, we will add a clarifying comment underneath as well stating that $\beta$ is a regularization parameter.
> * We want to point out that the empirical evaluation spanned a total of 15 diverse problem settings (5 prediction, 10 control), which is notably above average. We believe the selected problem settings well represent the space of potential applications for our work, while avoiding needless computational costs associated with e.g. Atari.

---

> > ### Comment · Reviewer_VsHs · 2021-12-01
> > **Response**
> >
> > Thank you for the rebuttal, I believe your response adequately address several of the issues I raised so I have given higher scores where appropriate.

---

### Official Review · Reviewer_9FeJ · 2021-11-02

**Correctness:** 4
**Technical Novelty And Significance:** 2
**Empirical Novelty And Significance:** 2
**Recommendation:** 6
**Confidence:** 2

**Main Review:**

Strengths.  The main contribution of this paper is using Huber Bellman error instead of mean squared Bellman error, since using the conjugate function to solve double sampling has been proposed in [1].
Weakness. In the last paragraph of page 8, it said that it applied QRC without a target network. Would the performance of QRC become better and outperform QRC-Huber when combined with the target network?

**Summary Of The Paper:**

This paper proposed using Huber Bellman error to robustify the loss function in learning value function and proposed using conjugate function to avoid double sampling. It also conducted experiments to justify its algorithm.

**Summary Of The Review:**

This paper is novel in the sense that it cooperates Huber loss with value evaluation to robustify value evaluation. It also conducts experiments to justify this improvement. Therefore, I think this paper is marginally above the acceptance threshold.

---

> ### Author Response · Authors · 2021-11-10
> **Author response**
>
> ### Contribution
> We agree that using conjugates to solve the double sampling issue is not novel to this paper, there is a broad literature using this technique and we have given extensive credit in our literature review. Formulating the biconjugate of the Huber function as a constrained optimization is novel, to the best of our knowledge. Proving theoretically that the MHBE bounds the MHVE and is thus a sensible proxy-objective function is novel to this paper. The resultant optimization algorithms, insights about using the secondary variable for gradient methods to control the loss surface, and the convergence proofs are all novel to this paper.
>
> We additionally introduce a novel control algorithm and show that (a) it has significant performance improvements over baselines and (b) that it specifically is notably more stable than heuristic uses of the Huber loss in RL (i.e. DQN). While we agree that our main contribution and investigation is the loss function itself, we hope that these further contributions were not overlooked.
>
> ---
>
> ### Target networks
> We point out, first, that not using target networks is a strength of our method, not a weakness. Target networks are well-known to reduce the sample efficiency of an algorithm in favor of improving stability. We derive update rules that are stable by design, and so do not rely on target networks for this stability. We evaluate these stability claims directly in the distribution plots in Figure 4. The sample efficiency of our algorithm and baseline QRC (which similarly does not require target networks) can be clearly seen in Figure 5.
>
> We further extensively study the impact of target networks on QRC and our proposed algorithm in Appendix D.1 (Figure 9). As might be expected, the performance of semi-gradient DQN drops when target networks are not used due to its instability. Also as expected, the performance of the stable gradient-based methods degrades as the target network refresh rate is decreased (longer time between updates).
>
> ---
>
> ### Scoring
> The review gives a 2 for technical novelty. We have addressed the review’s sole concern for technical novelty and would appreciate if the reviewer considers our response and reconsiders their score as appropriate, or further discuss otherwise.
>
> The review gives a 2 for empirical novelty. The paper already addresses the review’s sole concern for empirical novelty. We appreciate if the reviewer would reconsider this scoring, or provide further discussion if needed.

---

### Official Review · Reviewer_xg5t · 2021-11-03

**Correctness:** 3
**Technical Novelty And Significance:** 2
**Empirical Novelty And Significance:** 2
**Recommendation:** 6
**Confidence:** 2

**Main Review:**

Overall, I think this paper is interesting and is well motivated. My main concern is about its novelty. The authors highlighted that their main contribution is the introduction of two novel robust losses which are reformulated based on the least absolute loss and the Huber loss. This leads to the problem: why specifically the two losses? It seems to me that the comments and the analysis of this paper may also apply to other convex Lipschitz losses. It seems to me that it is the Lipschitz constant of the loss that "change the solution quality". I would expect more comments in this regard in the paper.

In addition, as a minor comment, the presentation of the paper could be further improved. For instance, the abbreviations, MHBE and MABE, were used without being defined beforehand.

**Summary Of The Paper:**

In this paper, the authors studied robust losses for learning value functions in reinforcement learning. The main contribution of this paper lies in the development of two novel robust loss functions for reinforcement learning and a saddle point reformulation based on the Huber and the absolute Belleman error and the biconjugates.

**Summary Of The Review:**

See above.

---

> ### Author Response · Authors · 2021-11-10
> **Author response**
>
> ### Novelty
> Although our methodology is general and can be applied to other losses, this does not limit its novelty. Our introduction of two new convex losses, error bounds on the value error for one of these losses, four prediction algorithms to minimize the new losses, and a novel control algorithm with notably improved performance over baselines and empirical stability and sensitivity analysis are all more than sufficiently novel contributions.
>
> This analysis does not hold for any convex, Lipschitz loss. It is necessary that the loss is convex, lower semicontinuous, and proper for the Fenchel-Moreau theorem to hold. It is also necessary that the biconjugate of the loss is known, and suitable optimization algorithms exist for minimizing it. There are many motivations for choosing these two loss functions, the title of the paper strongly implies one such reason: statistical robustness. The absolute error and Huber error are both well-known to be robust to statistical outliers. This is motivated in Figure 2, for instance, where we designed 5 prediction tasks of varying statistical properties, showing that the robust losses often perform notably better than the mean squared baseline. A primary insight of the work is that constraining the function space of the dual variable allows recovering statistically robust losses, an insight that could likely be used to yield other interesting loss functions as future work.
>
> ---
>
> ### Clarity
> >  the abbreviations, MHBE and MABE, were used without being defined beforehand.
>
> We point out that these abbreviations were defined in the sentence immediately preceding their first use (middle of page 2) as well as in the abstract. To improve clarity, we will additionally include the acronyms directly next to their definitions.
> > To do so, we develop a unified perspective of the mean absolute BE, mean squared BE, and mean Huber BE [...]
>
> ---
>
> ### Scoring
> We note the review gives a 3 for correctness but provides no supporting evidence. We would appreciate if the reviewer could expand on this decision so we can address these issues, or otherwise reconsider the scoring.
>
> We argue that suggesting the paper could have evaluated other losses is insufficient support for giving a 2 for technical novelty. We believe we have addressed this issue in our response and would appreciate if the reviewer would reconsider this scoring.
>
> Similarly, the score of 2 for empirical novelty is unsubstantiated---the review makes no comment about empirical results. We point out that we provide
> * a thorough (and novel) analysis of the stability of our proposed algorithms alongside baselines DQN and QRC on 5 domains,
> * evaluate the performance of these algorithms on 10 domains,
> * and provide 4 novel prediction learning problems designed to specifically target the strengths and weaknesses of our proposed methods.
> * In the appendix, we provide 5 ablation studies of specific hyperparameters and design decisions.
> * All experiments use at minimum 100 random seeds per condition, appropriate statistical significance tests and confidence intervals are provided, and statistical assumptions are validated (with distributions reported in the paper).
> * Finally, we extensively investigate both the strengths and weaknesses of our proposed algorithms, even designing counterexample domains to highlight the limitations of our approach in order to provide a complete and scientific picture of our contribution.
>
> We would strongly appreciate if the reviewer could revisit this score, or provide justification for their scoring.

---

> > ### Comment · Reviewer_xg5t · 2021-11-10
> > **Further comments**
> >
> > Here are my responses to the authors' reply:
> >
> > First: as the title indicated, the paper is mainly about robust losses for learning value functions. However, the authors did not propose any new loss functions but investigated the two existing and well-known ones (the mean absolute deviation loss and the Huber loss), which were developed by robust statisticians, in reinforcement learning. Frankly speaking, I feel that the authors proposed this title purposely to catch readers' eyes, which apparently overclaimed their contribution. I understand the algorithmic contributions made by the authors but have a strong feeling that the authors were exaggerating their contributions. This was why I chose "some of the paper’s claims have minor issues. A few statements are not well-supported, or require small changes to be made correct".
> >
> > Second: Having said above, it is most natural to ask what makes the two losses special? This is the question I proposed in my original review. The authors responded that "this analysis does not hold for any convex, Lipschitz loss." However, BOTH the absolute deviation loss and the Huber loss studied in this paper are convex and Lipschitz.
> >
> > Third: About the empirical comparisons, it is definitely not fair to compare the proposed robust approach with the non-robust MSBE, which is based on the fragile least square criterion.
> >
> > Fourth: Let me repeat that "the abbreviations, MHBE and MABE, were used without being defined beforehand". This was actually also pointed out by another reviewer. Other abbreviations used without definitions beforehand include the following ones: MHBE, MABE, QRC, QRC-Huber, DQN, RL, GTD2. I stopped trying to find out more, so this may not be a full list.

---

> > > ### Author Response · Authors · 2021-11-10
> > > **Response**
> > >
> > > We kindly request that the review and resulting discussion remain about the paper, not about opinionated takes on the authors or the authors' perceived intentions. We believe the purpose of this discussion period is to make sure we collectively understand the paper, its contributions, and whether it is fit for publication.
> > >
> > > It is no secret that the Huber function and absolute deviation are well-established loss functions and the paper certainly does not claim that these are novel to this work. However, their use in RL is **not** trivial due to the well-known double sampling issue. Likewise, note that it took nearly 20 years of development to derive algorithms to minimize the mean squared Bellman error in RL without bias due to double sampling. This is highly discussed in our work and a major contribution of our paper is deriving unbiased algorithms for two new Bellman errors in RL. We recommend reviewing Chapter 11 of the Sutton and Barto introductory textbook for further understanding of the challenges presented by double sampling.
> > >
> > > We hope that clarifying the challenges due to double sampling has alleviated this misunderstanding of our contribution and clarified that we in no way claim to be inventing Huber losses (invented by _Huber_) or the absolute deviation.
> > >
> > > What about the title insufficiently captures what this paper does? How would the reviewer propose modifying the title? Are there missing citations which would have made our contributions more clear?
> > >
> > > ---
> > >
> > > We apologize for the misunderstanding. When we said: "this analysis does not hold for any convex, Lipschitz loss" we intended to communicate that the analysis does not hold for *all* convex, Lipschitz losses. (I.e. one cannot select just any convex, Lipschitz loss and this analysis will hold.) We see how this wording was ambiguous.
> > >
> > > To further clarify, then, because the mean absolute and Huber functions are Lipschitz and convex, does not imply that all Lipschitz and convex functions will work in our analysis. By metaphor, just because a particular matrix $\mathbf{A}$ is symmetric does not imply that all matrices equal their transpose. We noted above that the restrictions are specifically that our function is (a) lower semi-continuous, (b) proper, and (c) convex. It is further useful if the biconjugate of the function is known, and if we have optimization algorithms which can minimize that biconjugate. That the function is Lipschitz is not directly relevant in our analysis.
> > >
> > > We further noted both in our submission and in our response that statistical robustness to outliers is what "makes the two losses special". We go so far as to empirically validate the utility of this on several hand-designed domains to highlight exactly this feature of the losses. Is this empirical investigation insufficient motivation for the proposed losses in the reviewer's opinion? And if so, why?
> > >
> > > We also provide a detailed and intuitive example in paragraph 4 of the paper, discussing intuitively when this robustness property is important in RL.
> > >
> > > ---
> > >
> > > Why is it unfair to compare a newly proposed loss to prior losses? Is there an empirical investigation that the reviewer would have preferred to see?
> > >
> > > We do not simply just compare the loss functions, their fixed-points, and the corresponding optimization algorithms. We specifically design problem settings where:
> > > 1. The prior work (the MSBE) has the best performing fixed-point, demonstrating the limitations of our own work
> > > 2. The proposed MHBE fixed-point performs best among the other losses.
> > > 3. The proposed MABE fixed-point performs best among the other losses.
> > >
> > > We strongly believe that we have done our due scientific diligence with these investigations and would appreciate if the reviewer could more concretely state what they find insufficient, and why.
> > >
> > > ---
> > >
> > > Perhaps we have a different definition of what it means to define an abbreviation, we believe using the full word in the sentence preceding the abbreviation is sufficient context. With that in mind, we have already stated in our previous response that we would adapt the writing by making the abbreviation appear immediately next to the word to reduce confusion. Was this response insufficient to the reviewer? What would the reviewer prefer we do instead?
> > >
> > > * QRC is defined via citation where it is first used. It is an algorithm name and so the acronym itself is unimportant.
> > > * QRC-Huber is a modification of that abbreviation where the modification is fully spelled out.
> > > * DQN is defined via citation **and** is an incredibly well-established acronym in the community. It is an algorithm name.
> > > * We will spell out "reinforcement learning" before using the acronym RL in the text.
> > > * GTD2 is defined via citation. It is an algorithm name.

---

> > > > ### Comment · Reviewer_xg5t · 2021-11-10
> > > > **Last response**
> > > >
> > > > First of all, let me clarify that I do not know who are the authors of this paper and nor have any opinionated takes on the paper.
> > > >
> > > > Here are my last responses:
> > > >
> > > > 1. As noted previously, the novelty of this paper is my major concern. Based on my reading of the paper and my own understanding, the contribution made in this paper is limited. I understand that this is a disappointing statement, but it is my non-opinionated comment.
> > > >
> > > > 2. I agree that the specific analysis conducted in this paper may not hold for all Lipschitz loss functions. However, I do believe that it is the Lipschitzness property of the two losses that help deliver robustness to the proposed approaches.
> > > >
> > > > 3. I think the title is inappropriate as the authors did not provide any theoretical characterizations of the properties that a general robust loss function needs to possess in the context of robust reinforcement learning but investigated two special robust losses. However, the current title may imply that the present study is on general robust loss functions for estimating valued functions. In addition, as mentioned by the authors, their major contribution is the algorithmic aspect, which also does not agree with the title.
> > > >
> > > > 4. "Why is it unfair to compare a newly proposed loss to prior losses?" Again, I do not agree that the two losses are "newly proposed losses". And it is definitely not fair/convincing to compare robust approaches with non-robust approaches.
> > > >
> > > > 5. About the abbreviations: I don't think this needs to respond further when two reviewers pointed out the same problem. Too many abbreviations without explicit definitions simply ruin the readability of the paper. For the sake of readability, there is nothing to argue with here.

---

> > > > > ### Author Response · Authors · 2021-11-10
> > > > > **Response**
> > > > >
> > > > > 1. If novelty is a concern, and this is an non-opinionated take, then it should be easy for the reviewer to justify these claims with a citation. Could the reviewer provide a citation where a mean absolute **Bellman error** or mean Huber **Bellman error** has been proposed in the literature? How does that paper avoid the bias due to double sampling, or ensure that these losses are learnable/identifiable? Does that paper produce online learning algorithms for minimizing those losses?
> > > > > 2. This is no longer a review of the paper, nor a relevant point. How does the Lipschitz-ness of the function change whether this paper should be published?
> > > > > 3. This isn't clear. This paper *is* about two robust loss functions for learning value functions. Would a better title have been: "Two robust losses for learning value functions"? Is this a sufficient criterion to prevent publication? Why?
> > > > > 4. If these are not newly proposed, then the reviewer can support their claims with a citation. If no such citation can be provided, then we posit that this is indeed a novel work and that the current scores do not appropriately reflect the submission. We also laid out in detail that we **do not simply compare these losses** in our empirical evaluation. The reviewer's comment of "fairness" simply makes no sense with regards to our evaluation strategy, and unfortunately, this final response provides no further justification or clarification on the matter, nor does it directly respond to our comments and questions.
> > > > > 5. We have already (repeatedly) stated that we would change the text in this regard. Is this a sufficient criterion to prevent publication? Why?
> > > > >
> > > > > We want to thank the reviewer for providing more details about their initial review and engaging in this discussion. We believe that the scores in the review are largely driven from a misunderstanding of the difficulty of developing loss functions whose gradients can be efficiently sampled in reinforcement learning. Due to this misunderstanding, it is clear that our paper might appear not novel (i.e. this prevents understanding _the most important_ contributions of our work) because the reviewer is out-of-field. We hope that the reviewer and meta-reviewer take this into account in their final scoring of this work.

---

### Author Response · Authors · 2021-11-20
**Updated submission**

Hello all!

We have uploaded an updated version of the paper. Specifically, we have:
* Ensured that the acronyms MHBE and MABE are defined directly next to their spelled-out versions (as opposed to in the immediately following sentence) to reduce confusion.
* Added a paragraph much earlier in the paper (top of page 5) discussing the relationship between our work and TDRC/QRC from Ghiassian et al. 2020. Discussion relating our contributions to TDC and GTD2 follow at the bottom of page 5.
* Pointed out that DQN is using a clipped loss, similar to a Huber loss, in the main body instead of the appendix.
* Modified the proof of Theorem 3.2 so that we are no longer using a vectorized version of the Huber function to maintain consistency.
* And modified subscripts on update rules throughout the paper to ensure consistency.

We believe we have addressed all of the actionable feedback that we have received from the reviewers at this point, and are happy to discuss any remaining questions, feedback, or confusions that may exist. The deadline for us to make further modifications to the paper is Nov 22nd.

---

### Decision · Program_Chairs · 2022-01-20

**Decision:**

Reject

**Comment:**

The paper proposes to use the Huber and absolute loss for value function estimation in reinforcement learning, and optimizes it by leveraging a recent primal-dual formulation by Dai et al.

This is a controversial paper. On one hand, it is a well motivated idea to apply robust loss on RL; the paper implemented the idea well by leveraging the saddle point formulation, and empirically demonstrate its advantages in practice.

On the other hand, the technical novelty of this paper is limited. The idea of Huber and standard conjugate formulation are straightforward application of existing techniques (despite being well motivated).

The authors seem to think that there has been no application of Huber loss on RL. But existing implementations of RL already uses Huber loss. For example, in the openAI baselines (https://openai.com/blog/openai-baselines-dqn/), they said the following:

"Double check your interpretations of papers: In the DQN Nature paper the authors write: “We also found it helpful to clip the error term from the update [...] to be between -1 and 1.”. There are two ways to interpret this statement — clip the objective, or clip the multiplicative term when computing gradient. The former seems more natural, but it causes the gradient to be zero on transitions with high error, which leads to suboptimal performance, as found in one DQN implementation. The latter is correct and has a simple mathematical interpretation — Huber Loss. You can spot bugs like these by checking that the gradients appear as you expect — this can be easily done within TensorFlow by using compute_gradients."

The authors discussed the first approach above on in the rebuttal, but I am not sure if the authors have considered the second method. If not, it would be worthwhile to discuss and compare with it.

See also "Agarwal et al. An Optimistic Perspective on Offline Reinforcement Learning" and "Dabney et al. Distributional Reinforcement Learning with Quantile Regression."

On the other hand, I have not seen the application of saddle point approach by primal-dual method of Dai on Huber specially.

It seems that the proposed algorithm is in the end equivalent to MSBE+primal-dual+ (h with softmax output). If it is that simple, I think it would help the readers to explicitly point this out upfront in the beginning (which is an interesting conceptual connection).  Because the primal-dual approach need to be approximate h with a neural network, the difference of the two methods is vague in the primal-dual space.

A side mark:  when we say "an objective for which we can obtain *unbiased* sample gradients", i think that the gradient estimator of the augmented Lagrange is unbiased; the gradient estimates of MHBE and MABE are still biased.

Overall, it is a paper with a well motivated and valuable contribution, but limited in terms of technical depth and novelty.